# When Splitting Makes Stronger: A Theoretical and Empirical Analysis of Divide-and-Conquer Prompting in LLMs

**Yizhou Zhang**[1,†]**, Defu Cao**[2]**, Lun Du**[3,‡]**, Qiang Fu**[4]**, Yan Liu**[2]
[1]Variational AI    [2]University of Southern California
[3]Ant Research    [4]Microsoft Research
{zyizhou96,dulun2834}@gmail.com, qifu@microsoft.com,
{defucao,yanliu.cs}@usc.edu

## Abstract

Foundation models, particularly Large Language Models (LLMs), have garnered significant interest due to their wide range of applications. Yet these models demonstrate notable weaknesses when confronted with tasks involving iterative sub-problems or deliberately misleading content—exemplified by complex arithmetic operations and comprehensive fake news evaluation. Conventional instructional prompting frequently produces flawed outputs in these scenarios. While research has established that advanced techniques such as Chain-of-Thoughts and Least-to-Most methodologies can dramatically enhance LLM performance, emerging investigation indicates that a more streamlined divide-and-conquer (DaC) approach—which systematically partitions input sequences into discrete components—can yield remarkable improvements for particular problem classes like misinformation assessment. Our investigation rigorously examines the efficacy of DaC prompting strategies and precisely delineates the task characteristics that benefit most from this methodology. Through comprehensive theoretical analysis, we establish formal guarantees for performance enhancement in specifically identified task categories. We validate our theoretical framework through focused empirical studies on large integer multiplication and factual verification tasks, where experimental outcomes robustly confirm our analytical predictions, demonstrating DaC's practical superiority in these challenging domains.

## 1 Introduction

Large language models (LLM) based on the Transformer architecture have led to major breakthroughs in natural language processing and other related fields in artificial intelligence (Jiang et al., 2025; Brown et al., 2020; Radford et al.; Touvron et al., 2023; Cao et al., 2024b). State-of-the-art general-purpose language models have demonstrated remarkable advancements in various domains, including question answering, graph learning, reading comprehension, text generation, and machine translation (Darm et al., 2025; Cao et al., 2024a; Chen et al., 2024; Tan et al., 2023; Hendy et al., 2023; Mao et al., 2023; Zong & Krishnamachari, 2023; Jia et al., 2024; Zhang et al., 2023a). Their remarkable adaptability stems not only from architectural innovations but also from increasingly effective prompting strategies that unlock their reasoning capabilities. These developments paves the way towards general-purpose problem solvers (Li et al., 2024; Bubeck et al., 2023).

However, as pointed out in (Wei et al., 2022), significant challenges emerge when applying even the most advanced LLMs to tasks involving extended solution paths, particularly those requiring sophisticated mathematical reasoning or complex knowledge integration. A series theoretic works attribute this challenge to **Parallelism Tradeoff** (Merrill & Sabharwal, 2023), a fundamental limitation of Transformers. Specifically, unlike Recurrent Neural Network, whose computational depth is linear to the input sequence length (i.e., the depth is $O(n)$, where $n$ is the input sequence length), Transformer does not contain any recurrent structure. Such design, while achieving superior parallelizability than

---

[†]Yizhou Zhang's work was done during his PhD program in the University of Southern California and internship in Microsoft.

[‡]Lun Du's work was done when he previously worked in Microsoft.

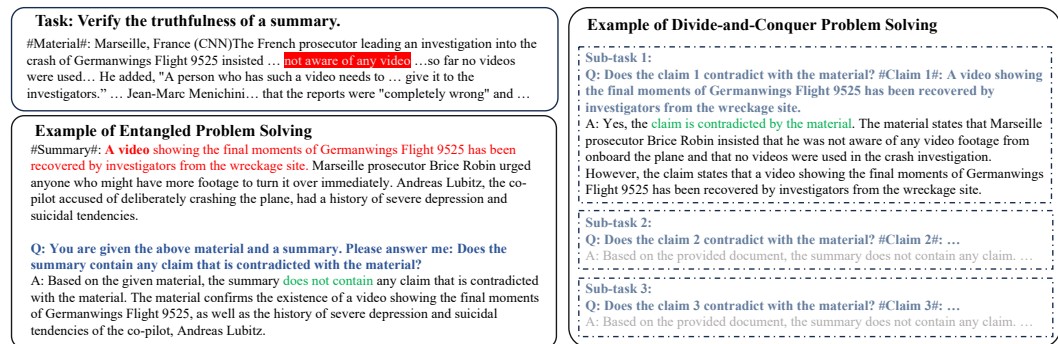

Figure 1: An illustrative example of hallucination detection with entangled problem solving (i.e., directly forward all inputs into the LLM) and divide-and-conquer problem solving (i.e., divide the problem inputs into parallel sub-tasks and tackle them in parallel). The sentence marked with red font in the material is the evidence that contradicts the first claim in the summary, which is marked with red font.

RNN, makes Transformers suffer from limited expressive power. Merrill & Sabharwal proved that the expressive power of fixed-depth log-precision Transformer, which is very close to the most commonly applied Transformer architecture for LLMs, is bounded by constant-depth logspace-uniform threshold circuits. This theoretical ceiling explains why these models struggle with tasks requiring extended sequential reasoning or precise tracking of long dependency chains, despite their impressive performance on many natural language tasks.

To address this challenge, carefully designed prompting strategies have been developed to tackle tasks that require stronger expressive power (Wang et al., 2025; Feng et al., 2023). A series of works focus on prompting the LLM with instructions or context samples to output the intermediate steps that derive the final answer in an autoregressive manner, such as Chain-of-Thoughts (CoT) (Wei et al., 2022; Wang et al., 2022; Zhou et al.; Chen et al., 2023). Some works further apply programs to guide LLM to strictly follow designated reasoning steps (Yao et al., 2023). Theoretically, these prompting strategies convert the role of the Transformer from a complete problem solver to a sub-problem solver in a dynamic programming or tree searching algorithm (Peng et al., 2024; Merrill & Sabharwal, 2024). In this way, these prompting strategies expand the expressive power of the LLMs and successfully improve the reasoning and searching of LLMs (Feng et al., 2023; Ye et al., 2024).

In contrast to methods that employ instructions, contextual examples, or programs to decompose reasoning processes into sequential intermediate steps, researchers have discovered that for certain tasks, LLM performance can be significantly enhanced through a fundamentally different approach: dividing input sequences into multiple sub-inputs and then merging the responses generated for each sub-input, as illustrated in Fig. 1. For example, Cui et al. proposes that in automated evaluation, LLM's performance can be further boosted by first dividing the input text into sentences and then evaluating them one by one. Intuitively, this paradigm benefits the tasks in a way similar to human brains, especially when the tasks are too hard or too complex. For example, when reviewing a long academic paper, some reviewers produce low-quality reviews (Garcia et al., 2021; Tennant & Ross-Hellauer, 2020; Cortes & Lawrence, 2021) containing hallucination-like **intermediate errors**, such as pointing out some 'missing baselines' that have already been sufficiently discussed by authors. To avoid such mistakes, experienced reviewers usually think slowly (Kahneman, 2011) to follow a **Divide-and-Conquer** paradigm to handle this task. Specifically, they decompose the paper review as examinations of multiple central opinions and then retrieve corpus to verify them respectively.

However, unlike Chain-of-Thoughts, whose advance in expressive power is supported by theoretical analysis (Feng et al., 2023), the performance boost from the Divide-and-Conquer paradigm lacks rigorous theoretical support. This theoretical gap leaves practitioners without clear guidance regarding the specific conditions under which DaC prompting provides demonstrable advantages over alternative approaches. To tackle this challenge, in this paper, we aim to understand the utility of DaC prompting. More specifically, we attempt to answer the following two research questions:

1. **RQ1: Compared to straightforward instructional prompting, does DaC have theoretically guaranteed advantages similar as CoT and its variants?**
2. **RQ2: Compared CoT and its variants, what utility and limitations does DaC have?**

To answer these questions, we first provide a theoretical paradigm that can help us analyze how the divide-and-conquer strategy expands the expressive power of fixed-depth log-precision Transformer on a given task. In this way, we provide a framework that can provide a theoretical guarantee to the DaC paradigm in various tasks. In this way, we present some conditions under which DaC have advantages compared to other prompting strategies. We then empirically evaluate DaC prompting and representative baselines on tasks that satisfy the proposed conditions and are challenging to existing prompting strategies even on state-of-the-art LLMs: Large Integer Multiplication, Hallucination Detection, Article-level Fact Verification (Cheng & Zhang, 2023; Li et al., 2023a; Wadden et al., 2020; Hu et al., 2024; Wu et al., 2023). These tasks either require very long reasoning paths (e.g., large integer multiplication) or contain deceptive contents (e.g., hallucination detection and fact verification), making existing methods like Chain-of-Thought prompting prone to intermediate errors. Importantly, these empirical findings align precisely with our theoretical predictions, confirming that DaC's advantages manifest most strongly in tasks featuring independent subtasks prone to intermediate errors—exactly as our theoretical framework anticipates.

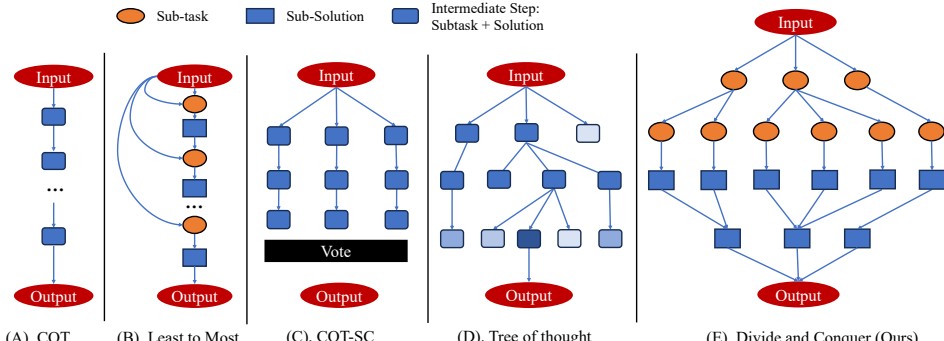

Figure 2: The comparison between DaC and the existing methods for prompting. The ellipse marks represent sub-tasks, the right-angled rectangles represent sub-task solutions, and the rounded rectangles represent intermediate steps that entangle sub-task and sub-solutions. The different shades in Tree of Thoughts (subfigure D) indicate the rates of different search directions. In CoT (Chain-of-Thoughts), CoT-SC, and ToT, the Large Language Models must simultaneously generate and resolve sub-tasks. Least-to-Most (also Decomposed Prompting) disentangles sub-task generation and resolution. However, its sub-task resolution and resolution assembly process are intertwined as it sequentially attaches new sub-tasks onto the previous resolution. Different from them, DaC totally disentangles the sub-task generation, sub-task resolution, and resolution assembly process.

## 2 Related Work

### 2.1 Expressive Power of Transformer

As discussed in previous works (Merrill & Sabharwal, 2023; Young, 2025; Feng et al., 2023; Yang et al., 2024), the expressive power of fixed-length log-precision transformers, which are widely applied in modern Pre-trained Large Language Models, is actually much more limited than people's expectations. Merrill & Sabharwal gives a theoretical proof that the expressive power of fixed-length log-precision transformers is upper-bounded with $\mathsf{TC}^0$. Feng et al. further extends their analysis to explain that a lot of common problems exceed the expressive power of fixed-length log-precision transformers. Such results explain why the powerful LLM may make some ridiculous mistakes and how CoT improves the performance.

### 2.2 Prompting Strategies of LLM

In this sub-section, we introduce the existing prompting and discuss their limitations and drawbacks. Following the notations in (Yao et al., 2023), we denote the Large Language Models with parameter

$\theta$ as $p_\theta$ and use lower case letters $x, y, z$ to denote input sequence, result, and intermediate steps, respectively.

**Input-Output (IO) Prompting** is the standard prompting strategy that attach input $x$ with instructions and/or few-shot in-context-learning examples to aqcuaire a prompt, denoted as $\mathsf{prompt}(x)$ (Yao et al., 2023). The LLM takes $\mathsf{prompt}(x)$ as input and predict result, i.e. $y \sim p_\theta(y|\mathsf{prompt}(x))$.

**Chain-of-Thought (CoT) Prompting** (Wei et al., 2022) aims at simulating humans' thinking process that handles complicated tasks (e.g., combinational reasoning and mathematical calculation) in a step-by-step manner. More specifically, the LLM is guided to output a series of intermediate steps $z_1, z_2, ..., z_n$ (also known as *thoughts*) autoregressively, i.e. $z_i \sim p_\theta(z_i|\mathsf{prompt}(x), z_1, ..., z_{i-1})$. Then the LLM output the prediction of result $y$ based on the *thoughts*, i.e. $y \sim p_\theta(z_i|\mathsf{prompt}(x), z_1, ..., z_n)$.

**Exploration-of-Thought (EoT) Prompting** and **Program-guided Prompting** are two variants of CoT. EoT includes a series of CoT's variants, such as Self-consistency with CoT (CoT-SC), prompting (Wang et al., 2022) and Tree-of-Thoughts (ToT), prompting (Yao et al., 2023), which aim at addressing the limitation of CoT in exploration. Their common central idea is to generate multiple chains of thought through sampling or proposing prompting, and then ensembling them to acquire a final prediction. Program-guided Prompting aims at controlling the LLM's generation process with symbolic programs or pre-defined procedures (Zhu et al., 2023; Jung et al., 2022; Zhou et al.; Khot et al., 2023; Creswell & Shanahan, 2022; Gao et al., 2023). Among them, the Least-to-Most (LtM) Prompting (Zhou et al.) and Decomposed Prompting (Khot et al., 2023) are close to this work. They are the earliest attempts that explicitly prompt the LLM to decompose the task as a series of sub-tasks and sequentially tackle them. LtM prompts an LLM to iteratively raise sub-tasks and sequentially solve them to acquire the final resolution. Decomposed Prompting can be regarded as an upgraded version of LTM. It introduces special notations into the prompt to represent program states and thus can call itself (i.e., recursion) or other modules (i.e., hierarchical decomposition), endowing it with stronger expressive power. Such a design increased the compositional generalization ability of LLMs in different areas, such as symbolic manipulation and multi-hop QA (Khot et al., 2023).

The aforementioned CoT and EoT families incorporate LLM with stronger expressive power than IO prompting. However, a critical issue of them is that, they could miss or ignore some important intermediate steps or contents (Paeng & Kwon, 2024; Liu et al., 2023). This problem is even worse when we are handling tasks involving long input (e.g., long documents and large numbers). Typical examples include large number arithmetic calculations and fact verification in long documents. Compared to them, Least-to-Most prompting and Decomposed Prompting introduce explicit task decomposition to enumerate sub-tasks. However, their task decomposers are based on multi-round conversation or question-answering, which navigate the LLM through the deceptive content's flow sequentially, and propagate the hallucination/deception in the contexts (Dziri et al., 2024; Yang & Ettinger, 2023), leading to decreased performance.

## 3 Preliminary of Divide-and-Conquer Prompting

In this section, we summarize and formalize **Divide-and-Conquer prompting strategy**. Unlike Tree of Thoughts (ToT), which interweaves reasoning with exploratory search, Divide-and-Conquer prompting strategy consists of three distinct stages: task decomposition stage, sub-task resolution stage, solution merge stage. In task decomposition stage, the LLM is prompted to explicitly decompose the task as a series of parallel homogeneous sub-tasks with smaller problem sizes (e.g. divide a long paragraph to sentences). Such design avoids the multi-round conversation or question-answering in LLM and Decomposed Prompting, making the model less prone to deception. After that, in sub-task resolution stage, the LLM is prompted to provide the solutions for every sub-task. Finally, in the solution merge stage, the LLM is prompted to assembly the solutions of subtasks and acquire the final answer. To tackle tasks of different sizes, Divide-and-Conquer prompting strategy can be divided to two variants: Single-Level DaC Solver and Multi-Level DaC Solver. Note that DaC requires additional tokens for sub-task decomposition and solution merging, but it reduces the average decoding context window size during sub-task resolution.

Single-level Divide-and-Conquer Solver decomposes the task in one call to the LLM, which expands the original task as a tree of one level. The algorithm is presented in the Alg. 1. The advantage of this variant is its simplicity and efficiency. However, when the original input is too long, single-level Divide-and-Conquer Solver may acquire sub-tasks with large problem sizes that will still trigger intermediate errors. In such a case, following (Khot et al., 2023), we can recursively expand the

---

**Algorithm 1** Single-Level Divide-and-Conquer Solver $T(S, a, t, L, f)$

---

**Require:** Input Sequence $S$, Prompt $m$ (for solution merge), Prompt $t$ (for sub-task tackling), Prompt $d$ (for task decomposition), LLM $L$
**Ensure:** Results of the task on input sequence $S$
  1: $\{S_1, S_2, ..., S_k\} \leftarrow L(d, S)$
  2: Result $\leftarrow \varnothing$
  3: **for** $i = 1, 2, ..., k$ **do**
  4:     Result $\leftarrow$ Result $+ [SEP] + L(t, S_i)$
  5: **end for**
  6: **Return** $L(m, Result)$

---

task as a multi-level tree. More specifically, we repeat the aforementioned steps to further divide the sub-tasks hierarchically until they are easy enough to be handled by the LLM. This can be done through a recursive program as presented in Alg. 2. More discussions on the proposed method's application scope, including its comparison with other prompting strategies and limitations, can be found in A.1.

---

**Algorithm 2** Multi-Level Divide-and-Conquer Solver Recursion $T(S, m, t, d, f, n, L)$

---

**Require:** Input Sequence $S$, Problem Size Metric Function $f(\cdot)$ (a function that measure the problem size), hyper-parameter $w$, Prompt $m$ (for merge), Prompt $t$ (for sub-task tackling), Prompt $d$ (for task decomposition), Large Language Model $L$
**Ensure:** Results of the task on input sequence $S$
  1: $S_1, S_2, ..., S_k \leftarrow L(d, S)$
  2: Result $\leftarrow \varnothing$
  3: **for** $i = 1, 2, ..., k$ **do**
  4:     **if** $f(S_i) > w$ **then**
  5:         Result $\leftarrow$ Result $+ [SEP] + T(S_i, m, t, d, f, w, L)$
  6:     **else**
  7:         Result $\leftarrow$ Result $+ [SEP] + L(t, S_i)$
  8:     **end if**
  9: **end for**
10: **Return** $L(m, Result)$

---

# 4 Theoretic Analysis

In this section, we provide theoretic analysis to the utility and limitations of the Divide-and-Conquer prompting. In the first subsection, we provide a comparison of IO prompting (common fixed-length instructional prompting) and DaC prompting in expressive power perspective. This part answers the first research question: the expressive power of IO prompting is a subset of DaC prompting. In the second subsection, we provide a comparison between Chain-of-Thoughts and DaC prompting in expressive power. Our comparison suggests that, although the expressive power of DaC prompting is a subset of Chain-of-Thoughts, for tasks satisfying specific conditions, DaC prompting can solve the problem with lower average context window length when decoding the tokens. Such property is empirically proved to be helpful for reducing the intermediate error and thus boost the performance.

## 4.1 Divide-and-Conquer vs. IO Prompting

We show that the expressive power of Divide-and-Conquer is stronger than IO Prompting: We denote the set of problems that a fixed-precision transformer with fixed-length IO prompting can tackle as $S(IO)$. Similarly, we denote the set of problems that a fixed-precision transformer with DaC prompting can tackle as $S(DaC)$. Then we have the following results:

$$S(IO) \subset \mathsf{TC}^0 \subseteq \mathsf{NC}^1 \subseteq S(DaC) \tag{1}$$

---

The function $f(\cdot)$ and the hyper-parameter $w$ are task-dependent and are often determined empirically based on the capabilities of the specific LLM being used (e.g., the number of digits where an LLM's multiplication accuracy drops, or the length of a text segment beyond which verification quality degrades).

**Proof Sketch:** The conclusion that $S(IO) \subset \mathsf{TC}^0$ is a corollary of the main results in (Chiang et al., 2023). In this paper, we mainly focus on proving $\mathsf{NC}^1 \subseteq S(DaC)$. This is significant because it establishes a rigorous computational complexity boundary between these prompting paradigms. Specifically, we exploit 2-color Binary Subtree Isomorphism (2-BSI) problem for the proof. In (Jenner et al., 2003), 2-BSI problem is proved to be an $\mathsf{NC}^1$-complete problem. Its definition is: **2-color Binary Subtree Isomorphism problem** is that, given a pattern 2-colorbinary tree $t_p$ and a base 2-color binary tree $t_b$, a solver is required to judge whether the pattern tree is isomorphic to a sub-tree of $t_b$ In (Jenner et al., 2003), the authors pointed out that the encoding of the problem will influence the hardness of the problem. In this paper, we focus on pointer list encoding of 2-BSI. Detailed information about the pointer list encoding of 2-BSI can be found in Appendix. For pointer list encoding of 2-BSI, we have the following theorem:

There exists a log-precision transformer with fixed depth $L$ and hidden dimension $d$ that can solve the 2-BSI of any size with fixed-length prompt $m$ (for merge), $t$ (for sub-task tackling) and $d$ (for task decomposition). **Proof Sketch:** The detailed proof is provided in the Appendix A.2. Here we give a brief flow of the proof. To prove this theorem, we first show an algorithm that can solve the problem with divide-and-conquer strategy. Then we prove that there exists a log-precision transformer with fixed depth $L$ and hidden dimension $d$ that can express the modules in the algorithms with different but fixed-length prompts. In this way, we can prove the theorem.

With the above theorem, we can prove that $\mathsf{NC}^1 \subseteq S(DaC)$, which finishes the proof. With this theoretic results, we can answer the **RQ 1**:

*Compared to IO prompting, DaC have theoretically guaranteed advantages in expressive power.*

## 4.2 DaC vs. CoT

In this section, we compare Divide-and-Conquer with Chain-of-Thoughts in order to understand the utility and limitation of DaC prompting. The limitation of DaC prompting is that its expressive power is a subset of CoT prompting: We denote the set of problems that a fixed-precision transformer with DaC prompting can tackle as $S(DaC)$. Similarly, we denote the set of problems that a fixed-precision transformer with CoT prompting can tackle as $S(CoT)$ Then we have the following results:

$$S(DaC) \subseteq S(CoT) \tag{2}$$

The proof of this proposition is very straightforward. For any problem that DaC can solve, we can concatenate all outputs of LLM in dividing, tackling and merging as a sequence. Then we can prompt LLM with CoT to output this sequence. Thus, the problem set that DaC can resolve is a subset of CoT.

The limitation revealed by the above theorem shows that compared to CoT, the appliance scope of Divide-and-Conquer is limited. However, by analyzing the average decoding context window size, we show that on specific tasks, divide and conquer can reduce the problem complexity: **Decoding Context Window Size:** In auto-regressive decoding, each token is decoded from a window that covers all previous tokens. We denote the length of the window as the Decoding Context Window Size of the token. Suppose that a task contains $k$ sub-tasks, each of which does not rely on the answers of other sub-tasks. We define such sub-tasks as **parallel sub-tasks**. If an LLM tackle these sub-tasks sequentially with CoT, then the average decoding context window size of the sub-tasks' resolution will be $C + \frac{\sum_{i=1}^{k} r_i - 1}{2}$, where $r_i$ is the length of the response to the $i$-th sub-task and $C$ is the length of input context. If we tackle them parallely with DaC, then the average decoding context window size of the sub-tasks' resolution will be $C + \sum_{i=1}^{k} \frac{(r_i-1)^2}{2\sum_{j=1}^{k} r_j} < C + \frac{\sum_{i=1}^{k} r_i - 1}{2}$.

The above proposition shows that when task contains a large amount of **parallel sub-tasks**, DaC is more helpful for reducing the average decoding context window size than CoT. Existing works have empirically showed that long decoding context window will propagate intermediate errors and thus increase the probability of generating hallucination (Yang & Ettinger, 2023). Thus, we can acquire a

---

$\mathsf{TC}^0$ represents problems solvable by constant-depth threshold circuits and is often associated with tasks like multiplication. $\mathsf{NC}^1$ represents problems solvable by logarithmic-depth circuits, indicating high parallelizability. The known inclusion $\mathsf{TC}^0 \subseteq \mathsf{NC}^1$ is crucial as it formally places the expressive power of DaC prompting in a higher complexity class than standard I/O prompting.

conclusion that DaC is competetive on tasks that contain a large amount of **parallel sub-tasks** and are bothered by intermediate errors and hallucination. With these theoretic results, we can answer the **RQ 2**:

*Compared to CoT and its variants, DaC prompting's expressive power is weaker. However, on tasks containing a large amount of* **parallel sub-tasks**, *DaC is more helpful.*

### 4.3 Advantages of DaC

The above analysis answer the two research questions that we proposed. By summarizing these two answers, we can acquire the two conditions such that when a task simultaneously satisfied both conditions, DaC bring performance boost:

- **Condition 1:** *the task is harder than $S(IO)$, such as $\mathsf{TC}^0$-complete problems and $\mathsf{NC}^1$-complete problems.*
- **Condition 2:** *the task contains a large amount of parallel sub-tasks and is bothered by hallucinations or intermediate errors.*

In Tab. 1, we present some sample tasks that satisfied the conditions. Also, we list some tasks that typically do not satisfy the conditions. This is helpful for guiding prompt engineering. Details are provided in Appendix A.7.

## 5 Experiments

| Applicable Tasks | Non-Applicable Tasks |
|---|---|
| Integer Multiplication | Integer Addition |
| Fact Verification | Multi-round QA |
| Consistency Evaluation | Planning |

Table 1: We list some example tasks that satisfy the conditions and some tasks that do not satisfy the conditions.

Our experiments are specifically designed to test the boundaries identified in our theoretical analysis. We selected tasks that directly correspond to the conditions outlined in Section 4.3. For integer multiplication (a $\mathsf{TC}^0$-complete problem), we strategically chose operand sizes that push beyond the reliable capacity of standard prompting to verify the theoretical complexity boundaries. For fact verification, we constructed test cases that require parallel verification of multiple claims, directly testing DaC's advantage in handling independent subtasks. Our evaluation metrics - edit distance for arithmetic and F1/precision/recall for verification tasks - were chosen to directly measure the impact of reduced intermediate errors, which our theory predicts should be DaC's primary advantage.

### 5.1 Case 1: Long Integer Arithmetic

In this case, we consider two tasks in long integer arithmetic: **Multiplication**, which satisfy the conditions we proposed, and **Addition**, which does not satisfy the first condition . Our experiment results will show that DaC prompting brings a performance boost on multiplication and does not bring a boost on integer addition.

**Task Setup:** For this task, we randomly generated 200 pairs of 5-digit integers. We choose 5 for the digit length because according to previous works, ChatGPT-3.5 gets 0% accuracy on 4-digit multiplications (Cheng & Zhang, 2023), and ChatGPT-4 gets close to 0% accuracy on 5-digit multiplications (Yang et al., 2023). We evaluate the performance with Edit Distance (Marzal & Vidal, 1993; Schaeffer et al., 2023).

**Setup of baselines and DaC:** In this task, our baselines include IO prompting, Chain of Thought (CoT), CoT-SC, Least-to-Most (LtM), and Decomposed Prompting (DeP). Tree-of-Thoughts is not applicable. This is because that multiplication is deterministic calculation without requiring search in a tree. For DaC, we apply Multi-Level Divide-and-Conquer program-guided solver.

---

Multiplication is a $\mathsf{TC}^0$-complete problem and can be divided to multiple parallel sub-tasks, while Addition is in $S(IO)$Barcelo et al. (2023)

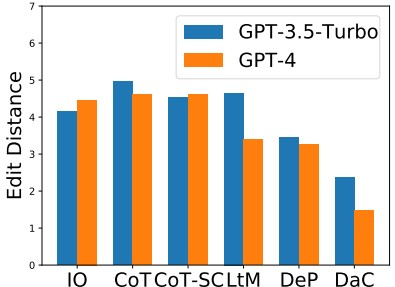
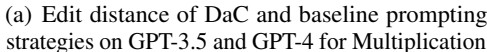
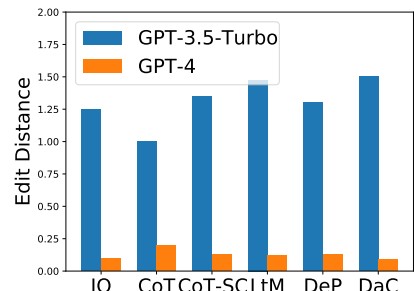

(a) Edit distance of DaC and baseline prompting strategies on GPT-3.5 and GPT-4 for Multiplication.

(b) Edit distance of DaC and baseline prompting strategies on GPT-3.5 and GPT-4 for Addition.

Figure 3: Performance of different prompting strategies on long integer multiplication.

| Strategies | GPT-3.5-Turbo | | | | GPT-4 | | | |
|---|---|---|---|---|---|---|---|---|
| | F1 | Acc | Prec | Recall | F1 | Acc | Prec | Recall |
| IO-prompting | 61.69 | 61.27 | 62.11 | 61.28 | 64.07 | 72.66 | **93.41** | 48.76 |
| Chain-of-Thoughts | 46.85 | 64.26 | **91.36** | 31.50 | 71.05 | 76.10 | 90.08 | 58.66 |
| CoT-SC | 47.70 | 64.25 | 88.83 | 32.60 | 71.39 | 76.36 | 90.41 | 58.98 |
| Tree-of-Thoughts | 70.40 | 59.91 | 55.83 | **95.34** | 69.41 | 71.73 | 75.53 | 64.28 |
| Least-to-Most | 56.43 | 64.91 | 74.42 | 45.44 | 72.51 | 77.11 | 90.74 | 60.38 |
| Divide-and-Conquer | **74.84** | **75.55** | 77.41 | 72.03 | **76.92** | **78.99** | 85.36 | **70.01** |

Table 2: Performance of different prompting methods on HaluEval dataset.

**Results:** Experimental results are shown in Fig. 3(a) and 3(b). As we can see, for integer addition which does not satisfy our proposed conditions, the performance of DaC, CoT and its variants does not significantly outperform IO prompting for both ChatGPT-3.5 and 4. However, for integer multiplication which satisfy our proposed conditions, under all settings, our proposed prompting strategy outperform all the baselines. This phenomenon indicate that our proposed conditions are useful for recognizing the tasks where DaC is more powerful.

## 5.2 Case 2: Fact Verification of Long Text

In the previous section, we show that for arithmetic tasks, our proposed conditions are discerning to the tasks where divide-and-conquer has advantages. In this section, we further present how our conditions can be applied to natural language tasks. Specifically, we present the performance of baselines and Divide-and-Conquer on fact verification of long text. In this task, the LLM is required to whether a long corpus is aligned with base knowledge. This task **satisfied the proposed two conditions**. For the first condition, we can reduce a 2-BTI problem to fact verification by describing the two trees with natural language. In this way, we can convert the trees to two paragraphs, and what we need to do is to ask the LLM to judge whether the two paragraphs are aligned or not. For the second condition, since we are tackling long text, each sentence can be regarded as a parallel sub-task. We select two benchmarks of fact verification: **Fact-Verification for Hallucination Detection** and **Fact-Verification for Misinformation Detection**

### 5.2.1 Hallucination Detection

Although Large Language Models have achieved impressive performance on various NLP tasks, they are bothered by hallucination problem (Manakul et al., 2023), especially when the generated content or the input context is too long for the user to have a thoroughly review (Zhang et al., 2023b). In this paper, we focus on evaluating the performance of different strategies in guiding LLM to recognize inconsistency between given context and model response with hallucination.

**Task Setup:** We use the HaluEval-Summary dataset. It is one of the three datasets in HaluEval benchmark for hallucination detection, which contains the hallucination generated by ChatGPT-3.5. HaluEval-Summary have the longest context and generated contents among all three tasks in this benchmark (Li et al., 2023a). Thus, detecting hallucination on this dataset requires repeatedly verify

| Strategies | GPT-3.5-Turbo | | | | GPT-4 | | | |
|---|---|---|---|---|---|---|---|---|
| | F1 | G-M | Prec | Recall | F1 | G-M | Prec | Recall |
| Io-Prompting | 72.12 | 72.77 | 83.22 | 63.64 | 69.15 | 71.77 | 94.44 | 54.55 |
| Chain-of-Thoughts | 56.09 | 60.64 | 90.48 | 40.64 | 74.03 | 75.79 | 94.21 | 60.96 |
| CoT-SC | 56.83 | 61.44 | **91.67** | 41.18 | 70.09 | 73.45 | **100.0** | 53.95 |
| Tree-of-Thoughts | 69.91 | 73.30 | 53.74 | **100.0** | 77.34 | 78.00 | 88.89 | 68.45 |
| Least-to-Most | 54.08 | 54.15 | 51.46 | 56.99 | 73.56 | 74.25 | 85.21 | 64.71 |
| Divide-and-Conquer | **76.88** | **77.13** | 83.65 | 71.12 | **81.11** | **81.24** | 76.67 | **86.10** |

Table 3: Performance of different prompting methods on SciFact dataset.

each sentence in the response, making standard prompting strategies acquire the worst accuracy across all three tasks. We report the Accuracy, F1 score (the hallucination pairs are positive samples), Precision and Recall.

**Setup of baselines, ablation variants and DaC:** In this task, our baselines include IO prompting, Chain of Thought, CoT-SC, Tree-of-Thoughts, Least-to-Most, and Decomposed Prompting. In this task, the sub-tasks are verifying fragments of the summary, which are homogeneous and do not require recurssion. In such a setting, Decomposed Prompting is equivalent to LtM. For this task, we apply single level Divide-and-Conquer solver to decompose the summary to multiple sentences, handle them separately and then merge the conclusions of all sentences. The details are in Appendix.

**Results:** Experimental results are shown in Tab. 2. For both GPT-3.5 and GPT-4, our proposed prompting strategy outperforms the baselines, presenting the advantage of DaC. More specifically, compared to IO-prompting, DaC achieved better performance in general, indicating the advantage brought by stronger expressive power. Meanwhile, compared to CoT and CoT-SC results, DaC clearly achieved much better recall. Tree-of-Thoughts, benefited by its searching ability, acquired a significantly better recall score compared to other baselines. However, its significantly lower precision substantially harms its overall performance and leads to accuracy that is even worse than standard IO-prompting. On the contrary, DaC carefully checked all sentences, located the one containing a factual error, and merged the answers. We hypothesize that this systematic and exhaustive approach contributes to DaC achieving a better balance between recall and precision, as it addresses each component without making overly broad or speculative inferences.

### 5.2.2 Misinformation Detection

The increasing abuse of misinformation toward manipulating public opinions on social media has been observed in different areas, such as healthcare (e.g. the recent COVID-19 pandemic) (Sharma et al., 2020; 2022). This threat is increasingly serious due to LLM's capacity in content generation (Li et al., 2023b; Weidinger et al., 2021; Zhang et al., 2022). This challenge raise the importance of fact-verification, which aims at judging the authenticity of an article based on a collection of evidence from verified source (Whitehouse et al., 2022; Zhang & Gao, 2023). In this experiment, we present that DaC can outperform other baselines in fact-verification involved with news article .

**Task Setup:** In this experiment, we mainly adopt SciFact dataset (Wadden et al., 2020). In SciFact dataset, each sample is a pair of news and evidence, where the evidence is the abstract of a peer-reviewed paper and the news is a sentence of claim. To better simulate the real-world scenario where news on social media usually appears as an paragraph of post, following Chen & Shu, we generate a dataset of paragraph-level misinformation based on SciFact dataset. Specifically, for a given claim, we apply ChatGPT-4 to extend the claim as an article based on the evidence. For this task, similar as hallucination detection, we apply single level Divide-and-Conquer solver to decompose the news article to multiple sentences, handle them separately and then merge the conclusions of all sentences. Also, the baselines in this experiments are the same as Hallucination Detection. The evaluation metrics includes F1 score, G-Mean score (geometric mean of precision and recall), Precision and Recall. We do not apply accuracy as the positive and negative classes are not balanced.

**Results:** Experimental results are shown in Tab. 3. Notably, GPT-3.5 incorporated with our proposed prompting strategy even outperform the performance of GPT-4 incorporated with IO-prompting, Least-to-Most, CoT and CoT-SC, which have significantly lower recall scores, indicating their proneness to deception. Only Tree-of-Thoughts, which is benefited by its advantage in exploring various options, acquired the best results among all baselines, but is still defeated by DaC. Moreover,

as we can see, for GPT-4 the performance of CoT-SC is even worse than CoT, which is supposed to be a specific case of CoT-SC without exploration. These results suggests that, when facing deceptive contents generated on purpose, existing works' improvement may not be robust.

# 6 Practical Guidelines for Practitioners

Based on our theoretical and empirical analyses, we offer practical guidelines for practitioners considering DaC prompting:

1. Task Assessment: DaC is most effective for tasks where: (a) subtasks can be processed independently, (b) the original task is prone to error accumulation, and (c) the task exceeds the reliable capacity of standard prompting.

2. Implementation Strategy: When implementing DaC, focus on clear subtask boundaries and explicit instructions for merging results. Our experiments show that template prompts that explicitly guide the model through decomposition, resolution, and merging phases yield the most reliable results.

3. Model Selection: While DaC provides benefits across different model capacities, the magnitude of improvement may vary. For critical applications requiring high precision, combining DaC with more capable models (e.g., GPT-4) provides the most reliable results, particularly for complex tasks.

4. Efficiency Considerations: For tasks with stringent latency requirements, DaC offers potential advantages through reduced context window size and opportunities for parallel processing, though this benefit must be weighed against the overhead of multiple model calls.

These guidelines provide a starting point for practitioners to effectively leverage DaC prompting in real-world applications.

# 7 Conclusions

In this paper, we analyze the utility and limitations of the divide-and-conquer prompting strategy. We first provide a theoretical analysis of Divide-and-Conquer prompting and compare it with representative prompting strategies. Based on these theoretical results, we summarize two conditions under which a task is suitable for Divide-and-Conquer prompting. After that, we conducted experiments on several tasks. The empirical results validated our theoretical analysis and show that the two conditions we proposed are helpful for recognizing the applicable scope of Divide-and-Conquer prompting.

# Acknowledgement

This work was supported in part by the Department of Defense under Cooperative Agreement Number W911NF-24-2-0133 and NSF Research Grant IIS-2226087. The views and conclusions contained in this document are those of the authors and should not be interpreted as representing the official policies, either expressed or implied, of the Army Research Office or the U.S. Government. The U.S. Government is authorized to reproduce and distribute reprints for Government purposes notwithstanding any copyright notation herein. D. Cao also acknowledges support from the USC Endowed Fellowship.

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

# A  Appendix

## A.1  Discussions and Limitations

In summary, the proposed method has the following advantages:

**Comparison with IO-Prompting: Superiority in Expressive Power** As we proved in Sec. 4, compared to IO-prompting, DaC has stronger expressive power and thus can solve harder problems.

**Comparison with CoT and EoT: Disentangling the task decomposition and task resolution** Compared to the prompting family of CoT and EoT, DaC explicitly separates the task decomposition stage and task resolution stage. Therefore, we can acquire explicit decomposed sub-tasks rather than intermediate thoughts proposed during decoding. Consequently, we can explicitly enumerate all sub-tasks output by the decomposition module and avoid the model from missing important sub-tasks.

**Comparison with CoT and ToT: Separates task decomposition from resolution**: Unlike Tree of Thoughts (ToT), which interweaves reasoning with exploratory search, DaC explicitly separates task decomposition from resolution, treating them as distinct phases. While ToT generates and evaluates reasoning paths simultaneously, DaC first identifies all relevant subtasks before solving any of them. Furthermore, DaC can handle heterogeneous decomposition patterns where subtasks vary in complexity through recursive application of the decomposition process, as shown in Algorithm 2. In cases where certain subtasks require deeper analysis, DaC recursively applies the decomposition until reaching manageable units, creating a multi-level tree structure with varying depths across branches.

**Comparison with LtM and Decomposed Prompting: Parallel Sub-task Handler and Sequential Sub-task Handler** Similar as DaC, some program-guided prompting like LtM and Decomposed Prompting also explicitly separate the task decomposition stage and task resolution stage. However, they are mainly designed for multi-step reasoning for complex tasks. Thus, they sequentially tackle the sub-tasks and assembly the resolutions. As a result, they tend to follow the flow of the deceptive contents, leading to proneness to deceptive content.

Although DaC surpasses the baselines on the proposed tasks, it still has some **limitations**. The first issue is that the appliance scope of DaC is still limited. More specifically, CoT, EoT, LtM and DaC are based on different algorithmic paradigms, learning to different Appliance Scopes. As pointed out by Feng et al., CoT and LtM can be considered as a neural **dynamic programming** algorithm. Thus, CoT is more suitable for tasks that can be bridged to dynamic programming, such as multi-step question answering. Differently, EoT is based on **exploration and search**, which is more suitable for planning and search, such as Game of 24 (Yao et al., 2023). DaC is based on **Divide-and-Conquer algorithm**. Thus, it is more suitable for tasks that can be decomposed to a series sub-tasks that are disjoint or only slightly overlapped. Our future work will focus on further expand the appliance scope of DaC to more areas like question answering.

## A.2  Proof to Theorem 4.2

Before providing the proof, we first formally define how to organize the inputs (i.e., two 2-color trees) as a sequence. We assume that we acquire two trees $t_p$ of size $n$ and $t_b$ of size $m$. They are organized as two sequences of nodes with a random order. Each node has three variables: color, left child index, and right child index. If any child is null, then the index is filled with 0. Then we can organize them as as two sequences $X_p \in \mathbb{R}^{n \times 3}$ and $X_b \in \mathbb{R}^{n' \times 3}$, where each item in the sequence is a vector of 3 dimensions. The first dimension is the index of the left child, the second dimension is the index of the right child, the third dimension is the color indicator (0 or 1). In addition, we have a root vector $r$ with three dimensions. The first dimension is the index of the root node of $t_p$ (i.e., pointing to the root node of $t_p$) and the second is the index of the root node of $t_b$ (i.e., pointing to the root node of $t_b$). The third dimension of $r$ is filled with 0 to make it have same dimension as the items in $X_p$ and $X_b$. This expression of trees is also called as pointer list encoding according to (Jenner et al., 2003). Note that in the following proof, we assume that all indices start from 1. Thus 0 is regarded as a NULL pointer.

Following the proof flow we provided in Sec. 4.2, we first provide the following divide-and-conquer algorithm that can solve the above problem:

---

**Algorithm 3** Recursion Divide-and-Conquer Algorithm for 2-BSI $BSI(r, X_p, X_b, m, t, d, f, w)$

---

**Require:** Inputs $r, X_p, X_b$, problem size metric function $f(\cdot)$, hyper-parameter $w$, merge function $m$, sub-task tackling function $t$, task decomposition function $d$

**Ensure:** A 0-1 indicator vector $v$: if there exists a subtree with node $i$ as root that is isomorphic with pattern tree $t_p$ defined with inputs $r, X_p, X_b$, then the $v[i]$ is 1. Otherwise, $v[i]$ is 0.

1: $r_l, r_r \leftarrow d(r, X_p, X_b)$
2: **for** $i \in \{l, r\}$ **do**
3:    **if** $f(r_i, X_p, X_b) > w$ **then**
4:       $v_i \leftarrow BSI(r_i, X_p, X_b, m, t, d, f, w)$
5:    **else**
6:       $v_i \leftarrow t(r_i, X_p, X_b)$
7:    **end if**
8: **end for**
9: **Return** $m(r, X_p, X_b, v_l, v_r)$

---

**Algorithm 4** Implementation of $d(r, X_p, X_b)$ when the depth of the tree indicated by $r$ is not longer than 2

---

**Require:** Inputs $r \in \mathbb{R}^3, X_p \in \mathbb{R}^{n \times 3}, X_b \in \mathbb{R}^{n' \times 3}$

**Ensure:** A 0-1 indicator vector $v$: if there exists a subtree with node $i$ as root that is isomorphic with pattern tree $t_p$ defined with inputs $r, X_p, X_b$, then the $v[i]$ is 1. Otherwise, $v[i]$ is 0.

1: $r_l \leftarrow\, < X_p[r[1], 2], r[2], r[3] >$
2: $r_r \leftarrow\, < X_p[r[1], 3], r[2], r[3] >$
3: **Return** $r_l, r_r$

---

**Algorithm 5** Implementation of $t(r, X_p, X_b)$ when the depth of the tree indicated by $r$ is not longer than 2

---

**Require:** Inputs $r \in \mathbb{R}^3, X_p \in \mathbb{R}^{n \times 3}, X_b \in \mathbb{R}^{n' \times 3}$

**Ensure:** A 0-1 indicator vector $v$: if there exists a subtree with node $i$ as root that is isomorphic with pattern tree $t_p$ defined with inputs $r, X_p, X_b$, then the $v[i]$ is 1. Otherwise, $v[i]$ is 0.

1: Initialize $v$ as all q vector with a length of $n'$
2: **if** $r[1] == 0$ **then**
3:    **Return** $v$
4: **end if**
5: **for** $i \in \{1, 2, ..., m\}$ **do**
6:    **if** $X_b[i, 3]\,! = X_p[r[1], 3]$ **then**
7:       $v[i] \leftarrow 0$
8:    **end if**
9: **end for**
10: **Return** $v$

---

The algorithm described above is a typical divide-and-conquer algorithm for solving rooted tree isomorphism. Its justification can be found in many textbooks introducing algorithms, such as *Introduction to Algorithms* (Cormen et al., 2022). Here we provide the detailed definition and implementation of problem size metric $f(\cdot)$, hyper-parameter $w$, merge function $m()$, sub-task tackling function $t(\cdot)$, task decomposition function $d(\cdot)$:

- $w = 1$, and $f(r, X_p, X_b)$ is defined as the depth of the pattern tree $t_p$ indicated with root vector $r$. Although precisely calculating $f(r, X_p, X_b)$ is of $O(n)$, judging whether $f(r, X_p, X_b) > 1$ only require us to check whether the root node has child. If not, then return False.

- $d(r, X_p, X_b) = r_l, r_r$ returns two new root vectors $r_l, r_r$. Both $r_l, r_r$ have the same second and third dimension as $r$. The $r_l$'s first dimension is updated to be the index of the left child of the root node that $r$ points to. The $r_r$'s first dimension is updated to be the index of the right child of the root node that $r$ points to. The updating function can be written as:

---

**Algorithm 6** Implementation of $m(r, X_p, X_b, v_l, v_r)$

---

**Require:** Inputs $r \in \mathbb{R}^3, X_p \in \mathbb{R}^{n \times 3}, X_b \in \mathbb{R}^{n' \times 3}, v_l \in \mathbb{R}^n, v_r \in \mathbb{R}^n$
**Ensure:** A 0-1 indicator vector $v$: if there exists a subtree with node $i$ as root that is isomorphic with
    pattern tree $t_p$ defined with inputs $r, X_p, X_b$, then the $v[i]$ is 1. Otherwise, $v[i]$ is 0.
1: Initialize $v$ as all 0 vector with a length of $n'$
2: **if** $r[1] == 0$ **then**
3:     **Return** $v$
4: **end if**
5: **for** $i \in \{1, 2, ..., m\}$ **do**
6:     **if** $X_b[i, 3] == X_p[r[1], 3]$ **then**
7:         **if** $v_l[X_b[i, 1]]] == 1$ and $v_r[X_b[i, 2]]] == 1$ **then**
8:             $v[i] \leftarrow 1$
9:         **else if** $v_l[X_b[i, 2]]] == 1$ and $v_r[X_b[i, 1]]] == 1$ **then**
10:            $v[i] \leftarrow 1$
11:         **end if**
12:     **end if**
13: **end for**
14: **Return** $v$

---

- $t(r, X_p, X_b) = v$ returns a 0-1 indicator vector $v \in \mathbb{R}^m$ with the same length of the base tree size. If there exists a subtree with node $i$ as root that is isomorphic with pattern tree $t_p$ defined with inputs $r, X_p, X_b$, then the $v[i]$ is 1. Otherwise, $v[i]$ is 0. When the pattern tree's depth is not higher than 1 (i.e., 1-node tree), $t(r, X_p, X_b)$ is equivalent to output a 0-1 vector indicating the nodes in the base tree that have the same color of the root node of pattern tree. The implementation is provided in Alg. 5.

- $m(r, X_p, X_b, v_l, v_l) = v$ merge the results $v_l, v_l$ to acquire a 0-1 indicator vector $v \in \mathbb{R}^m$ with the same length of the base tree size. If there exists a subtree with node $i$ as root that is isomorphic with pattern tree $t_p$ defined with inputs $r, X_p, X_b$, then the $v[i]$ is 1. Otherwise, $v[i]$ is 0. This function can be implemented by checking whether the pattern root's children have a perfect match with each node's children. Since each node has at most two children, checking the perfect match can be done in constant time. The implementation is provided in Alg. 6.

After providing the detailed implementation of the functions $d(\cdot), t(\cdot), m(\cdot)$, we are going to prove that there exists one unified transformer that can handle all these tasks with different prompts $d, t, m$. First, we will provide the following Lemma:

Any fixed-size logic circuit that only contains multi-fan-in AND gates, multi-fan-in OR gates, NOT gates and has no recurrent structure can be precisely simulated by a multi-layer perceptron (MLP) with ReLU activation function and a width of $O(|Input| + |Circuit|)$ and a depth of $O(|Circuit|)$, where $|Input|$ denotes the size of input and $|Circuit|$ denotes the number of gates in the circuit. Assume that we are given a series of input pins with logic variable of 0 or 1, organized as a 0-1 vector $x \in \mathbb{R}^h$. We first prove that all gates can be simulated by a two-layer perceptron. Then we can serialize all gates in the circuits and stack their corresponding 2-layer simulators accordingly to acquire a MLP simulator. An AND gate that take $x$ as input can be simulated as:

$$\text{AND}(x) = \sigma(w_A x - h + 1) \tag{3}$$

where $\sigma$ is the ReLU activation function, and $w_A$ is a weight vector with all dimensions equal to 1. If some dimensions of $x$ are not the input of the gate, we can set the corresponding dimensions in the weight vector as 0 and adjust the $h$ as the input pin number. Similarly, an OR gate that take $x$ as input can be simulated as:

$$\text{OR}(x) = 1 - \sigma(w_O x + h + 1) \tag{4}$$

where $\sigma$ is the ReLU activation function, and $w_O$ is a weight vector with all dimensions equal to -1. A NOT gate is different, since it only takes one input pin. In such a case, we denote the index of the input pin as $i$, then we can simulate a NOT gate as:

$$\text{NOT}(x) = \sigma(w_N x + 1) \tag{5}$$

where $w_N$ is a weight is a weight vector whose $i$-th dimension equals to -1 and all other dimensions equal to 0. Also, since the $x$ is a 0-1 vector, the activation function is equivalent to a identical function to $x$:

$$x = \sigma(x) \tag{6}$$

To construct a MLP that can simulate a fixed-size logic circuit without recurrent structure, we apply the circuit serialization in (Merrill & Sabharwal, 2024) which order the gates based on topological order. In this way, we can represent the circuit as a sequence GATE[1], GATE[2], GATE[3],...,GATE[L], where each GATE[i]'s input only contains the output of the previous gates and the original input $x$. Therefore, we can construct a $2L$-layer MLP base on the above serialization. Specifically, the $2i$-th and $2i + 1$-th layers of the MLP will simulate the $GATE[i]$ as well as copy all previous inputs with activation function and concatenate them together. This can be done by concatenate an identical matrix on the GATE's weight vector ($w_A$, $w_O$ or $w_N$). In this way, we can construct a MLP that precisely simulate the circuit. Since every time we concatenate the output of a gate with the input of it, the input dimension number of the final layer can be bounded by $O(|x| + L)$. In the worst case, for a circuit of size $L$, we needs $2L$ layers to precisely simulate it. However, in many cases, a lot of gates in the circuits can be run parallelly. In such cases, the MLP could be much more shallow.

Now, we can start to prove our main theorem:

There exists a log-precision transformer with fixed depth and hidden dimension that can solve the 2-BSI of any size with fixed-length prompt $m$ (for merge), $t$ (for sub-task tackling) and $d$ (for task decomposition). We prove this theorem by constructing a Transformer that can tackle this problem. First we define how to organize the input given $r$, $X_p$, $X_b$ and the prompt. Specifically, we construct a feature sequence $X \in \mathbb{R}^{(3+n+n')\times 7}$. Each item in this sequence is a feature of 7 dimensions, indicating a token. The first two dimensions indicate whether the token is a prompt ('00'), a root vector ('01'), a pattern tree node ('10'), or a base tree node ('11'). The third to fifth dimensions carries the information about the token. For a prompt token, '100' indicates merge prompt $m$, '010' indicates sub-task tackling prompt $t$, and '001' indicates task decomposition prompt $d$. For other cases, these three dimensions are with the same formula as the three dimensions in $r$, $X_p$, $X_b$. The rest two dimensions are allocated specifically for the merge function $m(\cdot)$ to store $v_l$ and $v_r$. More specifically, for the feature of token indicating the $i$-th base tree node, its sixth dimension is $v_l[i]$ and its seventh dimension is $v_r[i]$. For other tokens, these two dimensions are filled with 0. In $X[1]$ we store the prompt token. In $X[2]$ and $X[3]$ we store the input root vector $r$ duplicately. We store the same token twice so that we can tackle $r_l$ and $r_r$ separately. To separate this two token, we use the last dimension, which was padded as 0 in $r$, to distinguish them. $X[2,5]$ is set as 0 and $X[3,5]$ is set as 1. From $X[4]$ to $X[3 + n]$, we store $X_p$. From $X[4 + n]$ to $X[3 + n + n']$, we store $X_b$. For all node indices of pattern tree, we add them by 3. For all node indices of base tree, we add them by 3+n, so that the indices can be applied to directly retrieve the positional embeddings. After preparing the inputs, we start to construct our Transformer. The transformer first attach the position index for each token (positional embedding). After that, the inputs are forwarded into a transformer with depth of 2. Each transformer layer contains a multi-head attention layer followed by a MLP. As proved by (Merrill & Sabharwal, 2024; Feng et al., 2023), the attention layer of Transformer can retrieve the feature of tokens whose positional embeddings satisfy specific conditions. For multi-head attention, different heads can retrieve tokens with different conditions. In the following construction, we will use this conclusion to construct attention heads with different functions.

In the first Transformer layer, the function of each attention head is defined as:

- Head 1 only attends to the token itself to store $X[i]$ for token $i$.

- Head 2 attends to the token with a positional embedding matches the $X[i, 3]$ and copy this token's 5-dimension feature. For tree node tokens, this head's job is to retrieve the feature of $X[i]$'s left child. For root vector tokens, this head's job is to retrieve the feature of pattern tree root node. For the first token (prompt token), this head's retrieved feature will not be applied in the afterwards layers and thus does not influence the correctness of the model.

- Similar as Head 2, Head 3 attends to the token with a positional embedding matches the $X[i, 4]$ and copy this token's 5-dimension feature. This head's job is to retrieve the feature of $X[i]$'s right child. For root vector tokens, this head's job is to retrieve the feature of base tree root node.

---

**Algorithm 7** Logic circuit for MLP of the second Transformer layer

---

**Require:** Input feature $x'' \in \mathbb{R}^{42}$
**Ensure:** Output feature $y \in \mathbb{R}^7$
 1: $y \leftarrow x''[1:7]$ {Initialize $y$}
 2: **if** $x''[1:2] == 00$ or $x''[1:2] == 10$ {Prompt Token or Pattern Tree Node} **then**
 3:     **Return** $y$
 4: **else if** $x''[1:2] == 01$ {Root Vector Token} **then**
 5:     **if** $x''[24:26] == 001$ {Prompt is $d$} **then**
 6:         **if** $x''[5] == 0$ **then**
 7:             $y[3] \leftarrow x''[10]$ {get $r_l$, similar as line 1 in Alg. 4}
 8:         **else if** $x''[5] == 1$ **then**
 9:             $y[3] \leftarrow x''[11]$ {get $r_r$, similar as line 2 in Alg. 4}
10:         **end if**
11:     **end if**
12: **else if** $x''[1:2] == 11$ {Base Tree Node Token} **then**
13:     **if** $x''[24:26] == 010$ {Prompt is $t$} **then**
14:         **if** $x''[40] == x''[5]$ {Line 6 in Alg. 5} **then**
15:             $y[5] \leftarrow 1$
16:         **else**
17:             $y[5] \leftarrow 0$
18:         **end if**
19:     **else if** $x''[24:26] == 100$ {Prompt is $m$} **then**
20:         **if** $x''[13] == 1$ and $x''[21] == 1$ {Line 7 in Alg. 6} **then**
21:             $y[5] \leftarrow 1$
22:         **else if** $x''[14] == 1$ and $x''[20] == 1$ {Line 9 in Alg. 6} **then**
23:             $y[5] \leftarrow 1$
24:         **else**
25:             $y[5] \leftarrow 0$
26:         **end if**
27:     **end if**
28: **end if**

---

- Head 4 attends to the first token (prompt token) and copy this token's 7-dimension feature. This head's job is to retrieve the prompt indicator.

- Head 5 attends to the second token (root token) and copy this token's 7-dimension feature. This head's job is to retrieve the root information.

With the above 5 heads, the attention layer will output a 35-dimension feature for each token. We denote these features as $X' \in \mathbb{R}^{(3+n+n')\times 35}$. After that, these features are forwarded into a MLP fitting identical mapping to acquire the input features for the second Transformer layer.

In the second Transformer layer, the function of each attention head is defined as:

- Head 1 only attends to the token itself to store $X'[i]$ for token $i$.

- Head 2 attends to the token with a positional embedding matches the $X'[i, 31]$ and copy this token's 1-7 dimension features ($X'[X'[i, 31], 1:7]$). This head's job is to broadcast the feature of the pattern tree root node to every token.

With the above 2 heads, the attention layer will output a 42-dimension feature for each token. We denote these features as $X'' \in \mathbb{R}^{(3+n+n')\times 42}$. For root vector token, only the features from head 1 and head 4 are useful. For base tree node tokens, all 42 dimensions are useful. Then each token's feature are parallely forwarded into a MLP. We will use this MLP to fit the logical circuit described in Alg. 7. The function of Alg. 7 is to aggregate the functions of $m(\cdot), t(\cdot), d(\cdot)$ together and assign the correct value based on the prompt indicator. In Alg. 7, all operations are AND, OR, NOT, SELECTOR, and ASSIGN and there is not loop. Thus, it is a static logical circuit and can be implemented with multi-fan-in AND, OR, NOT gates. Thus, it can be precisely simulated by a MLP according to our Lemma A.2.

After acquiring the $y \in \mathbb{R}^7$ for each token, we can organize them as a feature sequence $Y \in \mathbb{R}^{(3+n+n')\times 7}$. When the prompt is $d$, we return $Y[2, 3 : 5]$ as $r_l$ and $Y[3, 3 : 5]$ as $r_r$. If the prompt is $t$ or $m$, then we can output $Y[3 + n + 1 : 3 + n + n', 5]$ as the expected $v$.

### A.2.1 Token Efficiency and Practical Trade-offs

While DaC requires additional tokens for sub-task decomposition and solution merging, it significantly reduces the average decoding context window size during sub-task resolution. The following illustrative tables demonstrate this trade-off.

Table 4: Illustrative Token Usage Comparison for $1234 \times 5678$

| Method | Stage | Est. Prompt Tokens | Est. Input Tokens | Est. Output Tokens | Stage Total Tokens |
|---|---|---|---|---|---|
| DaC (Single-Level) | 1. Decomposition | ~50 | (implicit) | ~25 | ~75 |
| | 2. Sub-Task Resolution | ~30 (each) | ~3 (each) | ~1-2 (each) | ~138 |
| | 3. Solution Merge | ~50 | (implicit) | ~2 | ~52 |
| DaC Total | | | | | ~265 |
| CoT | Full Task | ~25 | (implicit) | ~150 | ~175 |
| LtM | Full Task (Sequential) | ~90 | (implicit) | ~220 | ~310 |

Table 5: Illustrative Token Usage Comparison for Hallucination Detection (5-sentence summary)

| Method | Stage | Est. Prompt Tokens | Est. Input Data Tokens | Est. Output Tokens | Stage Total Tokens |
|---|---|---|---|---|---|
| DaC | 1. Decomposition | ~40 (instructions) | ~40 (summary) | ~70 | ~150 |
| | 2. Sub-Task Resolution | ~100 (base prompt) | ~165 (doc+statement) | ~10 (each) | ~1375 |
| | 3. Solution Merge | ~25 (instructions) | ~75 (analysis results) | ~10 | ~110 |
| DaC Total | | | | | ~1635 |
| CoT | Full Task | ~20 (instructions) | ~230 (doc+summary) | ~350 | ~600 |
| ToT (Illust.) | Combined Stages | ~50 (main prompt) | ~230 (doc+summary) | ~950 (all paths+eval) | ~1230 |

Reflecting on these illustrative comparisons, when considering the DaC prompting strategy, practitioners should weigh the potential increase in total token usage and the number of LLM calls against the significant benefits DaC can offer for specific task types. The tables suggest DaC might involve higher token counts, particularly if context is repeated across sub-tasks. However, this is a trade-off for key advantages, such as achieving superior performance and reliability for complex problems that align with its core principles. A crucial benefit of DaC is the reduction in the average decoding context window size for each sub-task, which can improve performance and manage complexity.

### A.3 Justification to Proposition 4.2

Suppose that the LLM is auto-regressively decoding $n$ tokens from an input context window with length of $C$. Then the decoding window of the $i$-th token is $C + i - 1$. Thus, the average window size will be:

$$\frac{\sum_{i=1}^{n}(C + i - 1)}{n} = \frac{C + n - 1}{2} \tag{7}$$

Thus, when we sequentially decode all the sub-task resolutions, the total length of the decoded sequence will be $\sum_{i=1}^{k} r_i$. Thus the average window size will be:

$$C + \frac{\sum_{i=1}^{k} r_i - 1}{2} \tag{8}$$

Meanwhile, when we apply Divide-and-Conquer, we parally decode each sub-task's resolution. Thus, for each sub-task, total window size will be $C \sum_{j=1}^{k} r_j + \sum_{i=1}^{k} \frac{(r_i-1)r_i}{2}$. Thus the average window size will be $C + \sum_{i=1}^{k} \frac{(r_i-1)r_i}{2\sum_{j=1}^{k} r_j}$. Meanwhile, with Jensen inequility, we have:

$$\sum_{i=1}^{k}(r_i - 1)r_i < \sum_{i=1}^{k}(r_i - 0.5)^2 \le \left(\sum_{i=1}^{k}(r_i - 0.5)\right)^2 \le \left(\sum_{i=1}^{k} r_i - 0.5k\right)^2 \tag{9}$$

Thus, when $k \ge 2$, we have:

$$\sum_{i=1}^{k}(r_i - 1)r_i < \left(\sum_{i=1}^{k} r_i - 1\right)^2 \tag{10}$$

Thus, we have:

$$C + \sum_{i=1}^{k} \frac{(r_i - 1)^2}{2\sum_{j=1}^{k} r_j} < C + \frac{\sum_{i=1}^{k} r_i - 1}{2} \tag{11}$$

## A.4 Refining Condition 2

To better illustrate Condition 2, we refine it into two distinct sub-conditions:

Condition 2.1 (Parallel Subtasks): The task must involve subtasks that can be processed independently and in parallel, without requiring intermediate results from other subtasks to proceed. Specifically, a task satisfies this condition if: (a) each subtask can be solved without relying on the outputs or reasoning paths of other subtasks, and (b) subtasks share a common structure or computational requirements, enabling consistent treatment by the model.

Condition 2.2 (Susceptibility to Intermediate Errors): The task must be prone to intermediate errors that arise during sequential reasoning. These errors typically originate from subtask solution errors—mistakes made during individual sub-task resolution—which then propagate through the reasoning chain and affect the final solution, creating a cascading effect that significantly impacts accuracy.

### A.4.1 Task Selection Rationale

Our task selection is driven primarily by scenario characteristics rather than granularity considerations. We deliberately selected tasks representing distinct types of dependencies between subtasks: Multi-round QA and Planning represent highly sequential reasoning where subtasks depend heavily on previous outputs, making them less suitable for DaC; while Verification and Consistency Evaluation involve naturally independent subtasks that align well with DaC's parallel processing capabilities. The difference in granularity between tasks is secondary to their structural characteristics—specifically, whether they exhibit the parallel subtask property (Condition 2.1) and susceptibility to intermediate errors (Condition 2.2) that our theoretical analysis identifies as crucial for DaC's effectiveness.

### A.4.2 Handling Heterogeneous Subtasks

In real-world applications, subtasks often vary significantly in complexity. For instance, in document analysis, some sections may contain dense technical content requiring further decomposition, while others might be straightforward. DaC's recursive nature naturally accommodates these variations by dynamically adjusting decomposition depth. When $f(Si) > w$ in Algorithm 2, the system recursively decomposes complex subtasks until reaching manageable units, creating branches of varying depths. This adaptive approach is particularly valuable for tasks like hierarchical summarization or technical document analysis where content complexity varies substantially throughout the input.

## A.5 Prompting Details of DaC

**Multiplication of Long Integers:** Suppose we have two $2n$-digit numbers $AB$ and $CD$, where $A, B, C, D$ are all $n$-digit numbers. Then we can break $AB \times CD$ as $(A \times C \times 10^{2n}) + (A \times D \times 10^n) + (B \times C \times 10^n) + (B \times D)$, where the calculation in each bracket pair is disjoint with others bracket pairs. We only need to compute the results of multiplication in each bracket pair parallelly and then merge all of them with addition:

*Decomposer Prompt $d$*: Please split the string a from the middle as two separated strings. The lengths of the two separated strings should be as close as possible. Please only return the two strings separated by a comma and do not return anything else.

*Sub-task Tackling Prompt $t$*: (1)Please compute $a * b$. (2) Please only return the final results and do not return anything else (ensure disentangled-sub-process principle).

*Merge Prompt $m$*: Please compute $x = a * 10^{2n} + b * 10^n$ and $y = c * 10^n + d$. Based on the above calculation, please compute $x + y$ carefully step by step.

**Long Integer Addition:** Similar to multiplication, we applied a divide-and-conquer approach to integer addition to empirically demonstrate its unsuitability. The prompts used were designed to

follow the same DaC structure for fair comparison, even though the task itself does not benefit from this paradigm. The prompts used were:

*Decomposer Prompt d:* "Please split the string a from the middle as two separated strings. The lengths of the two separated strings should be as close as possible. Please only return the two strings separated by a comma and do not return anything else."

*Sub-task Tackling Prompt t:* "Please compute the sum of the two numbers a and b. Please only return the final results and do not return anything else."

*Merge Prompt m:* "Please compute the final sum of the numbers a, b, c, and d. Please add them carefully step by step, ensuring you carry the values correctly from the rightmost digits. Show your work and return the final number."

**Hallucination Detection in Long Context**: We divide the summary to sentences. After that, we parallelly verify the sentences. Finally, we merge the verification to each sentence:

*Decomposer Prompt d*: Please help me segment the following paragraph as sentences. The separated sentence should be output as: #Statement 1#: ...#Statement 2#: ...Do not say anything else. Just return the statements in the given format.
Paragraph

*Sub-task Tackling Prompt t*: I want you to act as a factual contradiction checker. You are given a set of statements and a document. Among the statements, there might be one or more statements that contain contradictions with the document. Please find the problematic statement if it exist by analyzing the statements one by one. For each statement, please make a choice:

- A: The statement is totally aligned with the document for sure.
- B: The statement contradicts with the document.

*Merge Prompt m*: Based on the above analysis, please tell me, does any statement above contain contradiction with the document?.

**Fact-Verification for Misinformation Detection**: Similar as hallucination detection, we divide the summary to sentences. After that, we parallely verify the sentences. Finally, we merge the verification to each sentence. Thus, our decomposer prompt and sub-task tackling prompt are the same as hallucination detection. The only difference is the merge prompt.

*Merge Prompt m*: If we connect the above statements to be a news article, based on the above analyzation, please answer me: Is there any contradiction between the document and the article?

### A.6 Decomposed Prompting and Least to Most

Least-to-Most (LtM) Prompting (Zhou et al.) and Decomposed Prompting (Khot et al., 2023) are two similar works to our work. They both propose to explicitly prompt the LLM to decompose the task as a series of sub-tasks and sequentially tackle them. In Fig .2, we merge these two methods. Here, we will provide more detailed comparison of them, which is shown in Fig. 4. Decomposed Prompting can regarded as a upgraded version of LtM. It introduces special notations into the prompt to represent program states so that when sequentially tackling the sub-tasks, it can call heterogeneous modules to tackle them. Such design enable the LLM to call external programs (e.g., retrieval documents

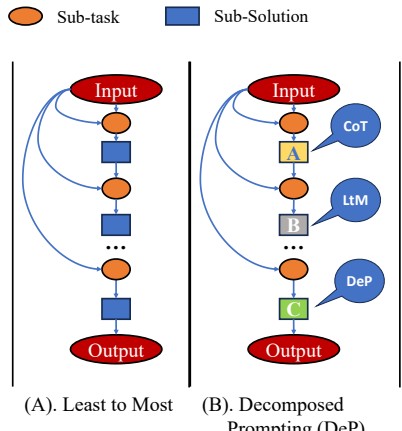

Figure 4: Comparison of Least-to-Most (LtM) Prompting and Decomposed Prompting (DeP).

on WikiPedia and program based calculator)
and/or itself (i.e., recursion). Such design
endows it stronger expressive power and increases the compositional generalization ability of LLMs
in different areas, such as symbolic manipulation and multi-hop QA (Khot et al., 2023). Also, it
endows LLM the ability to do open-domain QA by retrieving from external knowledge base.

### A.7 Typical Tasks that Satisfy and Dissatisfy the Proposed Conditions

To better assist the prompt engineering on different tasks, we list the typical tasks that satisfy and
dissatisfy the proposed conditions. In common tasks, the following tasks satisfy the proposed
conditions. For such tasks, searching good decomposition prompt for DaC is likely to be helpful for
the performance:

1. **Multiplication**
2. **Fact Verification on Long Text**
3. **Auto Evaluation on Long Text**
4. **Article-level Summary**

The following tasks typically do not satisfy the proposed conditions. For such tasks, searching good
decomposition prompt for DaC is not very likely to be helpful for the performance:

1. **Addition**: It is too simple and violate the condition 1
2. **Division**: It does not contain parallel sub-tasks, thus violate condition 2
3. **Multi-Round Question-Answering**: It is a typical sequential task, thus violate condition 2
4. **Planning**: It is a typical sequential task, thus violate condition 2

### A.8 More Discussions on Sequential Sub-task Tackling and Parallel Sub-task Tackling

**Example of Sequential Sub-task Tackling**
**Complete Task**: Compute 12345*67890:

**Sub-task 1: Compute x=45*90:**
A: ......

**Sub-task 2: Based on the above result, compute y=123*90*10^2+45*90:**
A: ......

**Sub-task 3: Based on the above result, compute z=45*678*10^2+123*90*10^2+45*90:**
A: ......

**Sub-task 4: Based on the above result, compute w=123*678*10^4+45*678*10^2+123*90*10^2+45*90:**
A: ......

**Example of Parallel Sub-task Tackling**
**Complete Task**: Compute 12345*67890:

**Sub-task 1: Compute x=45*90:**
A: ......

**Sub-task 2: Compute y=123*90*10^2:**
A: ......

**Sub-task 3: Compute z=45*678*10^2:**
A: ......

**Sub-task 4: Compute w=123*678*10^4:**
A: ......

**Resolution Assembly: Based on the above computation, compute x+y+z+w**
A: ......

Figure 5: Toy example of Sequential Sub-task Tackling and Parallel Sub-task Tackling in long integer
multiplication

Sequential Sub-task Tackling and Parallel Sub-task Tackling are two different paradigm in decom-
posing complex tasks as sub-task to tackle. The first one decompose a complex tasks as a series of
sub-tasks. In this series, each sub-task relies on the previous one's output as input or context. The
second one decompose a complex tasks as a set of sub-tasks, each of which does not rely on others.
Two examples for multiplication and hallucination detection are provided in Fig 5 and 6

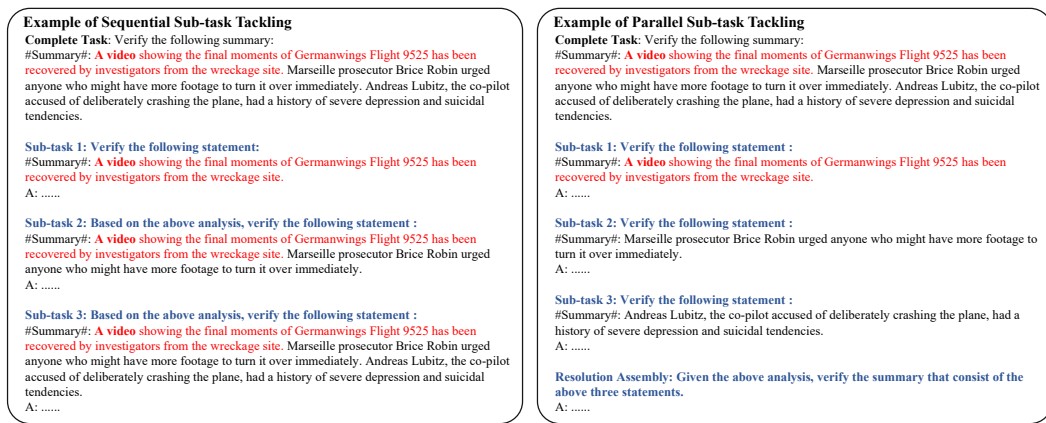

Figure 6: Toy example of Sequential Sub-task Tackling and Parallel Sub-task Tackling in hallucination detection

