# OpenReview forum: "When Splitting Makes Stronger: A Theoretical and Empirical Analysis of Divide-and-Conquer Prompting in LLMs"
_colmweb.org/COLM/2025/Conference — COLM 2025_

### Official Review · Reviewer_uHuS · 2025-05-05

**Rating:** 8
**Confidence:** 2
**Ethics Flag:** 1

**Summary:**

This paper is about a way of prompting language models (I use the word prompting with a large meaning) called Divide and Conquer. The paper first compares the approach with other methods like Chain of Thoughts. It then theoretically classifies the proposed approach according to equivalent computation models. This leads to a characterisation of two conditions on the task to solve, under which the proposed prompting method will be efficient. Finally the paper presents a list of experiment results on three types of tasks that confirm the the predicted behaviour.

**Clarity:** The logical flow of the paper is easy to understand.

**Originality:** I think that the paper is original in its way of handling and presenting the work. It presents an original way at analysing work on prompting. I was not aware of this kind of approach.

**Reliability:** I am not able to judge the well-foundedness and the exactness of the theoretical analysis, a field that I do not know. After documenting myself on the topic, I do not think that there is any reason to suspect any error.

**Significance:** Although the idea of the study is interesting, the results might not be so influencial. The characterisation in two conditions is interesting and somehow might sound paradoxical: some simple problems (large additions) will not be so well handled by the proposed prompting approach, while other simlarly simple ones (long multiplications) will profit from it.

**Questions To Authors:**

**Style:** I had problems with the syntax in many places. In particular subject-verb agreement is flawed in many different places. Because I am not a native speaker of English myself, I hate to say it, but correction using some automatic grammar-checker and spell-checker is really needed.

**Figure 1:** it takes time to identify the relevant parts #Material#, etc. in the figure and how they interact. A better layout might be welcome.

**Notations:** There are notations to explain IO prompting and CoT, but then no notation for the other prompting startegies. This is deceiving as the easy to understand formulae for the first two ones apear a lttle bit superfluous.

**Line 180:** finally, what are the criteria to decide when a sub-task "is easy enough to be handled by the LLM"?

**Theoretical Analysis:** A simple note on what TC^0 and NC^1 are would be welcome. Also the fact that it is known that TC^0 is included in NC^1 should be mentioned, and succintly explained, as this is missing frm the text.

**5.1. Longer Integer Arithmetic** I understand that the paper is following teh method in previosu wosk, but is edit distance the right way of measuring the success in the task. Shouldn't accuracy be preferred in the first place. Wouldn't the difference between the solution output by the LLM and the exact solution be a better metric?

**Table 2.** Some further analysis on the balance between recall and precision would be welcome. In particular it could be observed that the proposed approach is better balanced, wheras almost all other approaches are unbalanced and seem to favour either recall or precision. Is this due to some intrinnsic features of DaC?

**Location of tables** Tables 2 and 3 could be placed in the respective paragraphs they illustrate.

**Bibliography:** too many cited papers form ArXiv. Check whether these papers have been reviewed and published in reliable venues (I mean venues with a reviewing committee).

**Reasons To Accept:**

- Logical flow of the paper.
- Research questions are clearly identified.
- Theoretical analysis leading to clear conditions to delimit the efficiency of a proposed approach.
- Experimental results that confirm the predictions.
- For me, new way of considering how we should analyse prompting.

**Reasons To Reject:**

- Maybe too simple results.
- The theoretical results need some further explanation. and it might be questionable whether such a formalisation is really needed to extract the conditions.
- The algorithms are a little bit simplistic and might not be necessary.

The theoretical analysis is out of my competency. So, although I doumented myself and tried to understand, I cannot really judge. The opinion of the other reviewers is needed.

---

> ### Author Response · Authors · 2025-06-02
> **Response to Reviewer uHuS**
>
> Dear Reviewer uHuS,
>
> Thank you for your insightful and positive review. We are particularly encouraged by your appreciation for the paper's logical flow, the originality of the analytical approach to prompting, the clear identification of research questions, and the experimental confirmation of our theoretical predictions. We provide the following responses to your comments and questions:
>
> * **Results and Algorithms:** We believe the clarity and interpretability of the conditions derived from our analysis are a strength. Divide-and-Conquer (DaC) itself is a fundamental algorithmic paradigm, and our work demonstrates how its core principles, when applied to LLM prompting, lead to clear and significant improvements under well-defined circumstances. The algorithms (Alg. 1 & 2) are presented to formally specify the single-level and multi-level DaC processes, including recursion, which is crucial for the method's operationalization and understanding.
>
> * **Theoretical explanation:** We acknowledge that the theoretical analysis can be dense. The formalization was undertaken precisely to move beyond intuition and establish formal guarantees for DaC's performance enhancement—a point you noted as significant in your "Reasons To Accept." This formal approach also allowed us to rigorously define the boundaries (Conditions 1 & 2) where DaC is provably beneficial compared to standard prompting. We are happy to clarify any specific aspects further and, as detailed below, will add more explanatory notes for terms like $TC^0$ and $NC^1$.

---

> > ### Author Response · Authors · 2025-06-02
> > **Response to Questions of Reviewer uHus**
> >
> > We further provide the response to "Questions To Authors":
> >
> > * **Style (Syntax/Grammar):** Thank you for pointing this out. We will ensure the manuscript undergoes a thorough proofreading and correction process using grammar and spell-checking tools before the camera-ready version.
> > * **Figure 1 (Layout):** We appreciate this feedback. We will revise the layout of Figure 1 in the final version to more clearly delineate the components like #Material#, #Summary#, claims, and their interactions within the Entangled vs. Divide-and-Conquer problem-solving examples to enhance readability.
> > * **Notations (for other prompting strategies):** You are right that we provided detailed notations for Input-Output (IO) and Chain-of-Thought (CoT) prompting but not for all others. IO and CoT were formalized as they represent foundational baselines against which DaC's expressive power and theoretical advantages are directly compared (e.g., in Section 4.1 and 4.2). Other strategies like Tree-of-Thoughts (ToT), Least-to-Most (LtM), and Decomposed Prompting (DeP) are discussed more conceptually (e.g., in Figure 2, Section 2.2) to highlight their methodological differences from DaC. Introducing formal $p_{\theta}$ notations for every variant discussed would have significantly increased the notational load. We aimed for a balance, but we can consider adding a brief note if it clarifies comparisons.
> > * **Line 160 (Problem Size Metric & Threshold):** Your question about the criteria for deciding when a sub-task "is easy enough to be handled by the LLM" in Algorithm 2 is crucial for understanding the Multi-Level DaC Solver. This decision is governed by the "Problem Size Metric Function $f(\cdot)$" and the "hyper-parameter $w$". The function $f(S_i)$ measures the complexity of a sub-task $S_i$, and if this complexity exceeds the threshold $w$, the sub-task is further decomposed. An example is provided in Appendix A.3 for the 2-BSI problem: "$w=1$ and $f(r,X_p,X_b)$ is defined as the depth of the pattern tree $t_p$ indicated with root vector r.". For other tasks, $f(\cdot)$ and $w$ would be task-specific and often determined empirically based on the capabilities of the specific LLM being used (e.g., the number of digits where an LLM's multiplication accuracy drops, or the length of a text segment beyond which verification quality degrades). We will add a sentence in Section 3 to clarify that $f(\cdot)$ and $w$ are task-dependent and often empirically tuned for optimal performance.
> >
> > * **Theoretical Analysis ($TC^0, NC^1$ explanation):** This is a helpful suggestion to improve accessibility. In the camera-ready version, we will add a brief explanation of these complexity classes – for instance, $TC^0$ representing problems solvable by constant-depth threshold circuits (often associated with tasks like multiplication) and $NC^1$ representing problems solvable by logarithmic-depth circuits (indicating high parallelizability). We will also explicitly mention and briefly explain the known inclusion $TC^0 \subseteq NC^1$ (as used in our paper: $S(IO) \subset TC^0 \subseteq NC^1 \subseteq S(DaC)$).
> > * **Longer Integer Arithmetic (Edit Distance Metric):** We chose Edit Distance for the long integer arithmetic task as it is a standard metric for comparing sequences, and LLM outputs are sequences of tokens. It is widely used in evaluating generative tasks, including arithmetic, where the output is a string of digits. Edit Distance captures partial correctness, which is valuable when LLMs make minor errors (e.g., one or two incorrect digits) versus producing a completely different number. While accuracy (exact match) is indeed important, it can be an overly harsh metric for very long number arithmetic, where models might be close but not perfect, yielding 0% accuracy and thus less granularity in observing improvements. The numerical difference between the LLM's output and the exact solution is another valid metric. Our choice of edit distance was guided by its generality and its ability to penalize structural inaccuracies in the generated number string. We believe it provides a good measure of how "close" the LLM's output is to the correct sequence.

---

> > ### Author Response · Authors · 2025-06-02
> > **Continue Response to Questions of Reviewer uHus**
> >
> > * **Table 2 (Balance between Recall and Precision for DaC):** This is an excellent observation. While our current paper does not provide a direct theoretical proof for why DaC might achieve a better balance in Table 2 for hallucination detection, we hypothesize it stems from DaC's systematic and exhaustive approach. By decomposing the task into smaller, independent sub-tasks (e.g., verifying individual claims/sentences), DaC ensures each component is addressed. This focused attention on each sub-part might lead to fewer missed items (improving recall) without making overly broad or speculative inferences that could harm precision, as each sub-task resolution is relatively contained. The merging stage then aggregates these more reliable, focused judgments. In contrast, end-to-end methods like CoT might sometimes overlook details in a long context or reasoning chain (affecting recall) or make inferential leaps (affecting precision).
> >
> > * **Location of Tables (Tables 2 and 3):** We agree. In the camera-ready version, we will ensure that Tables 2 and 3 are placed as close as possible to their first mention or the relevant text that discusses them, to improve the flow and readability of the paper.
> > * **Bibliography:** We understand this concern. In a rapidly advancing field like LLM research, ArXiv is often the first venue for disseminating cutting-edge findings. For the camera-ready version, we will diligently review our bibliography and update ArXiv citations to their peer-reviewed published versions wherever they have become available.
> >
> > Thank you once again for your detailed and constructive feedback. Your comments and questions will certainly help us improve the clarity and comprehensiveness of our paper.

---

> > > ### Author Response · Authors · 2025-06-09
> > > **A Gentle Follow-up with uHuS**
> > >
> > > Dear Reviewer uHuS,
> > >
> > > We are writing to briefly follow up on our previous response to your review.
> > >
> > > We wanted to gently check if our clarifications have sufficiently addressed the valuable points you raised. We would be happy to answer any further questions you might have.
> > >
> > > Thank you once again for your positive support and for the insightful feedback on our work.

---

> > > > ### Comment · Reviewer_uHuS · 2025-06-09
> > > >
> > > > Thank you for the answers to the questions. I understand that the published manuscript will contain the relevant modifications if it is accepted, which is the wish expressed by my score.

---

### Official Review · Reviewer_kq9V · 2025-05-11

**Rating:** 7
**Confidence:** 4
**Ethics Flag:** 1

**Summary:**

The paper proposes the divide-and-conquer (DaC) prompting strategy and identifies two conditions under which it leads to performance gains. It provides a theoretical framework to determine the types of tasks where DaC prompting is advantageous and offers a theoretical guarantee of its effectiveness. The authors empirically validate their analysis on tasks like Large Integer Multiplication and Hallucination Detection, tasks that involve long reasoning paths.

**Questions To Authors:**

Can you include a token usage comparison between the 3 methods?

**Reasons To Accept:**

- The theoretical analysis on the Divide-and-Conquer prompting method is solid and well-motivated
- The performance gains obtained on the 2 tasks presented are encouraging

**Reasons To Reject:**

- Unless a token usage comparison is available between DaC, CoT, ToT, the adoption of DaC may not be as broad as it involve overheads in decomposition and merging steps

---

> ### Author Response · Authors · 2025-06-02
> **Response to Reviewer kq9V**
>
> Dear Reviewer kq9V,
>
> We sincerely thank you for your positive assessment of our paper, particularly for recognizing the solid and well-motivated theoretical analysis and the encouraging performance gains from the Divide-and-Conquer (DaC) prompting strategy.
> While our paper notes that "DaC requires additional tokens for sub-task decomposition and solution merging, but reduces the average decoding context window size during sub-task resolution" and discusses efficiency trade-offs in Appendix A.2, we agree that concrete comparisons would strengthen our analysis. To further address your valuable suggestion and illustrate these trade-offs more concretely, we've prepared some illustrative token usage comparisons for tasks discussed in our paper:
>
> Table 1. Illustrative Token Usage Comparison for $1234 \times 5678
> | Method                 | Stage                       | Estimated Prompt Tokens | Estimated Input Tokens | Estimated Output Tokens | Stage Total Tokens |
> | :--------------------- | :-------------------------- | :---------------------- | :--------------------- | :---------------------- | :----------------- |
> | **DaC (Single-Level)** | **1. Decomposition** | ~50                     | (implicit in prompt)   | ~25                     | **~75** |
> |                        | **2. Sub-Task Resolution** | ~30 (each)              | ~3 (each)              | ~1-2 (each)             | **~138** |
> |                        | **3. Solution Merge** | ~50                     | (implicit in prompt)   | ~2                      | **~52** |
> | **DaC Total** |                             |                         |                        |                         | **~265** |
> | **CoT** | **Full Task** | ~25                     | (implicit in prompt)   | ~150                    | **~175** |
> | **LtM**| **Full Task (Sequential)** | ~90                     | (implicit in prompt)   | ~220                    | **~310** |
>
> Table 2. Illustrative Token Usage Comparison for Hallucination Detection (5-sentence summary)
>
>
> | Method          | Stage                      | Est. Prompt Tokens (excl. main data) | Est. Input Data Tokens | Est. Output Tokens | Stage Total Tokens |
> | :-------------- | :------------------------- | :----------------------------------- | :--------------------- | :----------------- | :----------------- |
> | **DaC** | **1. Decomposition** | ~40 (instructions)                   | ~40 (summary)          | ~70                | **~150** |
> |                 | **2. Sub-Task Resolution** | ~100 (base prompt each)              | ~165 (doc+statement each) | ~10 (each)         | **~1375** |
> |                 | **3. Solution Merge** | ~25 (instructions)                   | ~75 (analysis results) | ~10                | **~110** |
> | **DaC Total** |                            |                                      |                        |                    | **~1635** |
> | **CoT** | **Full Task** | ~20 (instructions)                   | ~230 (doc+summary)     | ~350               | **~600** |
> | **ToT (Illust.)**| **Combined Stages** | ~50 (main prompt instructions)       | ~230 (doc+summary)     | ~950 (all paths+eval) | **~1230** |
>
>
> Reflecting on these illustrative comparisons, when considering the DaC prompting strategy, practitioners should weigh the potential increase in total token usage and the number of LLM calls against the significant benefits DaC can offer for specific task types, as identified in our paper. The tables suggest DaC might involve higher token counts, particularly if context like a source document is repeated across sub-tasks. However, this is a trade-off for key advantages. As discussed in our paper (e.g., Section 4.3), DaC is beneficial for tasks harder than $S(IO)$, contain many parallelizable sub-tasks, and are prone to intermediate errors or hallucination. Our paper's Appendix A.2 provides practical guidelines, emphasizing task assessment for suitability—DaC excels with independent subtasks where standard prompting might lead to error accumulation. A crucial benefit of DaC is the reduction in the average decoding context window size for each sub-task (as highlighted in your provided summary table and discussed in lines 212-220 of our paper), which can improve performance and manage complexity, especially for models with context limitations. Furthermore, the parallel nature of DaC sub-tasks offers potential for reduced latency. While CoT might appear more token-efficient for simpler tasks, DaC's structured approach can unlock superior performance and reliability for complex problems that align with its core principles, like isolating and preventing "deceptive content flow" in verification tasks (as mentioned in lines 137-139 of our paper regarding sequential methods). Thus, the potentially higher token cost can be a justified investment for these gains in reliability.
>
> We thank the reviewer for highlighting this practical aspect, and we will incorporate such discussion in our revised version.

---

> > ### Author Response · Authors · 2025-06-09
> > **A Gentle Follow-up with kq9V**
> >
> > Dear Reviewer kq9V,
> >
> > We are writing to briefly follow up on our previous response. We hope the illustrative token usage tables and the detailed analysis of the trade-offs of the DaC strategy were helpful in addressing your questions about practical overheads.
> >
> > We would be very grateful to know if this has sufficiently addressed your concerns. Please let us know if you have any further questions.
> >
> > Thank you again for your positive score and for providing feedback that will help us improve the paper.

---

> ### Comment · Reviewer_kq9V · 2025-06-10
>
> Thank you authors for providing these details! Including this in the final paper if accepted would be beneficial to the community. I will retain my positive score.

---

### Official Review · Reviewer_1hW4 · 2025-05-12

**Rating:** 8
**Confidence:** 4
**Ethics Flag:** 1

**Summary:**

This paper explores the DaC (Divide and Conquer) strategy for enhancing language model performance, particularly in solving tasks that can be decomposed into parallel subtasks. The approach is grounded in rigorous theoretical analysis, demonstrating DaC's ability to efficiently solve NC1-complete problems, while outperforming conventional methods like CoT (Chain of Thought) in tasks like multi-digit multiplication and hallucination detection. Practical templates and criteria for applying DaC further enhance its usability. However, the study is constrained by its focus on parallelizable tasks, lacking complexity analysis for multi-level DaC and offering a limited evaluation restricted to synthetic or curated datasets. A more comprehensive theoretical analysis and diverse empirical validation would strengthen the work's generalizability.

**Questions To Authors:**

None

**Reasons To Accept:**

1.	Rigorous theoretic support for DaC strategy. The paper provides a solid theoretical foundation for the DaC (Divide and Conquer) strategy, offering a clear computational complexity boundary that highlights DaC's advantages for specific problem classes. This foundational contribution is significant because previous studies lacked formal guarantees for DaC. Specifically, the paper rigorously proves that DaC can solve NC1-complete problems, such as the 2-color Binary Subtree Isomorphism, whereas IO prompting is limited to TC0 (as discussed in Section 4.1, Equation 1). This theoretical insight establishes DaC's superiority in addressing these computational challenges.
2.	Reduction of intermediate errors in parallelizable tasks. The DaC strategy demonstrates a clear advantage in minimizing intermediate errors during parallelizable tasks. The experimental results validate this advantage by showing that DaC effectively reduces error propagation in tasks with parallel subtasks (Condition 2), which significantly outperforms sequential methods like CoT (Chain of Thought). For instance, in the 5-digit multiplication task, DaC achieves a lower edit distance compared to CoT and IO prompting. Additionally, in hallucination detection, DaC shows a remarkable improvement, increasing the F1 score by approximately 10% over CoT on GPT-4. These empirical results are well-aligned with the theoretical predictions, reinforcing the practical utility of the DaC approach.
3.	DaC strategy is of good practical utility. The paper effectively bridges theory and practice by providing actionable guidelines for employing the DaC strategy. It introduces practical prompting templates, such as decomposition prompts for multiplication, and outlines clear criteria for identifying tasks that are well-suited to DaC. By explicitly separating task decomposition from resolution, DaC avoids the pitfalls of entangled reasoning, thereby enhancing clarity and efficiency. This practical guidance greatly contributes to the adoption of DaC in real-world applications.

**Reasons To Reject:**

1.	Limited task scope The DaC strategy is inherently restricted to tasks that can be effectively decomposed into parallel subtasks. As a result, it is unsuitable for sequential or interdependent problems, which violate the core assumption of parallelizable subtasks (Condition 2.1). This limitation is explicitly acknowledged in the paper, as evidenced by the exclusion of multi-round QA and planning tasks from Table 1. However, this narrow applicability reduces the generalizability of DaC. Furthermore, the paper does not explore potential hybrid approaches, such as combining DaC with CoT (Chain of Thought) for tasks that involve a mix of parallel and sequential reasoning, which could have broadened its applicability.
2.	Insufficient theoretic analysis. Although the paper provides a strong theoretical foundation for DaC’s performance on specific problem classes (e.g., Theorem 4.2 for 2-BSI), the theoretical analysis is incomplete. Specifically, the complexity of recursive DaC is not fully explored. While it is mentioned that a multi-level DaC solver may encounter subtasks with large problem sizes, the associated overhead is neither quantified nor analyzed. This omission is critical, as the absence of complexity analysis for multi-level decomposition (e.g., token overhead and computational cost) raises concerns about the scalability of DaC in real-world scenarios. A more comprehensive analysis would have greatly enhanced the theoretical depth of the work.
3.	Evaluation scope is limited. The empirical evaluation of DaC is primarily confined to controlled conditions, such as synthetic tasks (e.g., 5-digit multiplication) and curated datasets. While these experiments demonstrate DaC’s advantages under ideal settings, they fail to capture its performance in more diverse or challenging scenarios. Specifically, the paper does not test DaC on noisy inputs, multi-modal tasks (e.g., text combined with images), or adversarial examples, which are common in real-world applications. This limited evaluation reduces confidence in DaC’s generalizability and robustness. A broader range of experiments, including cross-domain generalization and stress-testing, would significantly strengthen the practical relevance of the proposed approach.

---

> ### Author Response · Authors · 2025-06-02
> **Response to Reviewer 1hW4**
>
> Dear Reviewer 1hW4,
>
> We are very grateful to the reviewer for their thorough assessment, positive evaluation, and insightful comments. We are particularly encouraged that the reviewer recognized the rigorous theoretical support for our Divide-and-Conquer (DaC) strategy, its effectiveness in reducing intermediate errors in parallelizable tasks, and its practical utility with actionable guidelines.
>
> We appreciate the suggestions for strengthening the work's generalizability and offer the following responses:
>
> 1. **Task scope:** The current paper's primary goal was to first establish the theoretical guarantees and empirical validation for "pure" DaC within its identified applicable domain.
> We acknowledge that the DaC strategy, as presented in this paper, is primarily designed for tasks that can be effectively decomposed into parallel subtasks (Condition 2.1). This focus was intentional to clearly establish its foundational principles and benefits for this specific class of problems. As the reviewer notes, Table 1 explicitly lists tasks like multi-round QA and planning as non-applicable due to their sequential nature, thereby defining the current scope. The suggestion to explore hybrid approaches, such as combining DaC with CoT for tasks involving mixed reasoning, is an excellent one. Indeed, our future work aims to "expand the appliance scope of DaC to more areas like question answering", which would naturally involve investigating such hybrid models.
>
> 2. **Further theoretical analysis** (for multi-level DaC complexity): Our primary theoretical focus was on establishing the expressive power bounds of DaC relative to other prompting strategies (Section 4.1, 4.2)  and identifying the conditions under which DaC is beneficial.
> We do, however, qualitatively address efficiency considerations. The paper notes that "DaC requires additional tokens for sub-task decomposition and solution merging, but it reduces the average decoding context window size during sub-task resolution" in #line 152. Furthermore, Appendix A.2, Practical Guidelines for Practitioners,  states, "For tasks with stringent latency requirements, DaC offers potential advantages through reduced context window size and opportunities for parallel processing, though this benefit must be weighed against the overhead of multiple model calls".
> To further illustrate how overheads might accrue in a multi-level scenario, we've worked through an illustrative token usage and call count estimation for a 5-digit multiplication, like 12345×67890, which is a task that would utilize the Multi-Level DaC Solver (Algorithm 2).
>
> Here’s a table summarizing the illustrative breakdown:
>
>
> Tabel 1: Illustrative Multi-Level DaC for $12345 \times 67890$
> | Effective Problem Solved by DaC                      | Requires Further Recursion? | Est. LLM Calls for this Problem's DaC Steps (Decomp, Sub-Solves, Merge) | Est. Tokens for these DaC Steps | Notes                                                                  |
> | :--------------------------------------------------- | :-------------------------- | :------------------------------------------------------------------------- | :---------------------------------- | :--------------------------------------------------------------------- |
> | **Level 1: $12345 \times 67890$** | Yes (for 3 of its 4 sub-problems) | ~3 (L1 Decomp, 1 Direct Solve for $P_4$, L1 Merge)  *+ calls from Level 2* | ~138 (for L1 ops) *+ tokens from Level 2* | $P_4=45 \times 90$ solved directly. $P_1, P_2, P_3$ recurse to Level 2.          |
> | **Level 2: $P_1 = 123 \times 678$** | No                          | ~6 (Decomp, 4 Direct Solves, Merge)                                        | ~157                                | Base cases are 1/2-digit multiplications.                              |
> | **Level 2: $P_2 = 123 \times 90$** | No                          | ~6 (Decomp, 4 Direct Solves, Merge)                                        | ~157                                | Base cases are 1/2-digit multiplications.                              |
> | **Level 2: $P_3 = 45 \times 678$** | No                          | ~6 (Decomp, 4 Direct Solves, Merge)                                        | ~157                                | Base cases are 1/2-digit multiplications.                              |
> | **Overall Illustrative Totals** |                             | **~21 LLM Calls** | **~609 Tokens** | Sum of Level 1 direct ops and all Level 2 operations.                  |

---

> > ### Author Response · Authors · 2025-06-02
> > **Continue Response**
> >
> > This illustrative multi-level analysis for 5-digit multiplication aims to shed light on your valid concerns regarding the complexity of recursive DaC. The overhead primarily manifests as an **increase in the total number of LLM calls** (21 in our 2-level example for a single 5-digit multiplication) and the consequent **accumulation of tokens** across these calls (~609 in this illustration), which includes tokens for the decomposition and merging meta-prompts at each recursive step. This directly relates to your points about quantifying overhead and scalability. The depth of recursion, and thus the total overhead, is controlled by the Problem Size Metric Function $f(\cdot)$ and the hyper-parameter $w$ (from Algorithm 2), which determine when a sub-task is "easy enough" to be solved directly. While our paper's current theoretical focus was on establishing expressive power (e.g., Theorem 4.2 for 2-BSI) rather than providing a formal $O$-notation complexity for token/call scaling in recursive DaC for tasks like arithmetic, this example illustrates the mechanics. For an N-digit multiplication split into halves, the recursion depth would be $O(\log N)$, and the number of base operations (direct solves) would scale with $N$ raised to some exponent depending on the exact split (e.g., $N^{\log_2 3}$ for Karatsuba's 3 sub-problems). Here, each "operation" (decomposition, direct solve, merge) is an LLM call. The critical trade-off, as argued in our paper, is that this operational overhead is incurred to achieve higher accuracy and handle problem complexities that might otherwise lead to failure, by ensuring the LLM consistently operates on more manageable sub-problems, thereby reducing intermediate errors. A more comprehensive formal analysis of this recursive complexity, quantifying token overhead and computational cost more generally, is indeed a valuable direction for future theoretical work, building upon the foundations we've presented.
> >
> >
> > We hope this illustrative breakdown is helpful in further clarifying the practical implications of the multi-level DaC approach and demonstrates our careful consideration of your feedback.
> >
> > 3. **Evaluation scope**: We appreciate your observation regarding the types of tasks and datasets used in our current empirical evaluation. Our current evaluation, while focused, was deliberately designed to rigorously test DaC's core principles and boundaries by:
> >   - Including "bound case" tasks (e.g., large integer multiplication) where DaC is theoretically strong, validating its performance against complex reasoning and deceptive content.
> >   - Testing an "out-of-bound" task (integer addition) to empirically confirm DaC's predicted limitations and the applicability of our theoretical conditions.
> >   - Covering both human-guided decomposition strategies (e.g., for mathematical tasks) and LLM-automated decomposition (e.g., sentence splitting for text, detailed in Appendix A.6 ).
> >
> > This focused design was a deliberate choice to validate DaC's core advantages and the "boundaries identified in our theoretical analysis", particularly the reduction of intermediate errors, in settings where these effects could be clearly measured against our predictions.
> >
> > We concur with the reviewer that testing DaC's performance on a broader range of conditions, including noisy inputs, multi-modal tasks, and adversarial examples, represents important next steps. We view this as a natural and valuable extension of the current foundational study to assess wider real-world robustness and generalizability.

---

> > > ### Author Response · Authors · 2025-06-09
> > > **A Gentle Follow-up with 1hW4**
> > >
> > > Dear Reviewer 1hW4,
> > >
> > > We are writing to briefly follow up on our previous response to your review.
> > >
> > > We wanted to gently check if our clarifications have sufficiently addressed the valuable points you raised. We would be happy to answer any further questions you might have.
> > >
> > > Thank you once again for your positive support and for the insightful feedback on our work.

---

### Official Review · Reviewer_N6F9 · 2025-05-17

**Rating:** 7
**Confidence:** 2
**Ethics Flag:** 1

**Summary:**

The paper presents a prompting method for LLMs called Divide-and-Conquer (DaC). It splits the input sequence into several independent parts and then merges the outputs. Through theoretical analysis and experiments, the authors test the effectiveness of DaC and identify the task types for which DaC is suitable.

**Reasons To Accept:**

The paper builds a thorough theoretical framework for DaC prompting, compares it with previous methods such as IO prompting and CoT, and reveals the task categories where DaC is applicable. I must admit that fully understanding the theoretical analysis is beyond my expertise, so I cannot offer a complete assessment of their strengths and weaknesses. However, the analytical approach itself appears valuable and worth referencing.

**Reasons To Reject:**

- The paper does not provide detailed prompt templates for concrete problems. In the DaC process, is the task decomposition done by directly prompting the LLM to split the task itself, or is there human intervention?
    - If the task is split by humans, the decomposition strategies can vary widely and may be impractical in real applications, are there proposed solutions?
    - If the LLM itself decomposes the problem before answering each sub-question, this path may be more interesting. How are the prompts designed? Since there are many possible decomposition strategies, will there be incorrect splits? Can the LLM determine whether it is worth using the DaC strategy and decide to split the task on its own?
- How exactly is long integer addition handled with DaC, such that its performance becomes even worse than the previous baselines? Many details seem to be missing.

---

> ### Author Response · Authors · 2025-06-02
> **Response to Reviewer N6F9**
>
> Dear Reviewer N6F9,
>
> We sincerely thank you for your positive assessment of our theoretical framework and for your insightful questions regarding the practical implementation and nuances of the Divide-and-Conquer (DaC) prompting strategy. We appreciate the opportunity to clarify these aspects.
>
> ### Detailed Prompt Templates and Task Decomposition
>
> We would like to clarify that detailed prompt templates for the specific tasks used in our experiments (Large Integer Multiplication, Hallucination Detection, and Misinformation Detection) are provided in Appendix A.6 of the submitted manuscript. These include the exact prompts used for the task decomposition ($d$), sub-task tackling ($t$), and solution merging ($m$) stages. For instance, for Large Integer Multiplication, the decomposition prompt $d$ is: "Please split the string a from the middle as two separated strings. The lengths of the two separated strings should be as close as possible. Please only return the two strings separated...". Similarly, for text-based tasks like Hallucination Detection, the decomposer prompt $d$ guides the LLM to segment paragraphs into sentences.
>
> In our DaC framework, the LLM executes the task decomposition based on specific prompts ($L(d,S)$ as shown in Algorithms 1 and 2). The strategy for decomposition, however, is determined by the task type and the desired level of controllability. Our paper presents comprehensive experiments of decomposition that incorporate both human-guided and LLM-based approaches.
> - For numerical tasks like multiplication, the decomposition strategy is typically human-defined, following established algorithms (e.g., splitting numbers, as described for AB×CD). These strategies are then translated into prompts for the LLM to execute, ensuring precision and adherence to the algorithm. By carefully crafting the decomposition prompt $d$, we can direct the LLM to split tasks in a way that is most beneficial for the subsequent sub-task processing and merging stages.
> - For text-based tasks such as fact verification or consistency evaluation (examples of generation/analysis tasks), the decomposition is guided by prompts that instruct the LLM to segment the input into meaningful, parallelizable units, like sentences.
>   - Prompt Design for LLM Decomposition: Prompts are designed to guide the LLM toward a specific, useful decomposition suitable for parallel processing, as detailed in Appendix A.6.
>   - Handling Incorrect Splits: While incorrect splits by the LLM are possible, the Multi-Level DaC Solver (Algorithm 2) can recursively further decompose sub-tasks that remain too complex. We recognize that developing more sophisticated methods for detecting and correcting decomposition errors represents an important avenue for future research.
>   - LLM Autonomy in Choosing DaC: Currently, the decision to apply DaC is based on our theoretical analysis of task characteristics (Section 4.3). Extending the framework to enable autonomous LLM decision-making regarding DaC application presents a promising direction for future work, potentially enhancing both the flexibility and effectiveness of our approach.
>
> ### Long Integer Addition with DaC
>
> We thank the reviewer for this question, as it allows us to clarify an important aspect of our experimental design. As shown in Table 1, the Long Integer Addition task was intentionally chosen as a case where DaC is not expected to provide benefits, thereby helping to validate the boundaries of our proposed conditions for DaC's effectiveness. The following is the detailed explanation:
>
> - Theoretical Unsuitability: As stated in Section 5.1, integer addition "does not satisfy the first condition", which is that "the task is harder than S(IO)". Applying DaC to such tasks introduces overhead (multiple LLM calls for decomposition, sub-task solving, and merging) without a corresponding benefit in expressive power or error reduction for that specific task type.
> - Demonstrating Boundary Conditions: The purpose of including the addition was to empirically demonstrate that DaC does not universally improve performance and that its advantages are specific to tasks meeting the identified criteria. The results in Figure 3b (showing DaC performing worse than or comparable to IO prompting for addition) confirm our theoretical prediction that DaC is not advantageous for this type of problem. While the DaC mechanism (splitting numbers, processing parts, then summing) can be mechanically applied to addition, the inherent simplicity of addition relative to the LLM's capabilities means the decomposition and merging steps add unnecessary complexity and potential for minor errors or inefficiencies, leading to slightly worse performance.
>
> We believe these clarifications address the reviewer's concerns. We are confident in the contributions of our work, particularly the theoretical formalization and the identification of its applicable scope. We will incorporate these clarifications into the manuscript.

---

> > ### Comment · Reviewer_N6F9 · 2025-06-08
> >
> > Thank you for your response, which has resolved most of my concerns. Therefore, I am willing to raise my score from 6 to 7. Additionally, I have one minor question: I could not seem to find the detailed prompt design for the Long Integer Addition task. Since this may influence DaC's underperformance, I still recommend including these details in the final version.

---

> > > ### Author Response · Authors · 2025-06-09
> > > **Thanks for increasing the score**
> > >
> > > Dear Reviewer N6F9,
> > >
> > > Thank you very much for your positive feedback on our rebuttal and for raising your score. We greatly appreciate your constructive engagement and support for our work.
> > >
> > > Regarding your final question about the detailed prompts for the Long Integer Addition task, you have raised a very valid point. Our initial appendix focused on tasks where DaC is advantageous. We will add the specific prompts used for the Long-Integer Addition experiment to the appendix in the camera-ready version of our paper.
> > >
> > > Thank you again for your helpful suggestions, which will certainly improve the final version of our manuscript.

---

### Decision · Program_Chairs · 2025-07-08

**Decision:**

Accept

**Comment:**

The paper proposes a new method of prompting: divide-and-conquer, and provides theoretical understandings on under what conditions can this prompting help improve performance.

Pros:
- The paper is well-written.
- The theoretical analysis is insightful and provides groundings of whether the proposed prompting works vs not, and on what type of tasks.
- Experiments on a few real tasks showed the effectiveness (and ineffectiveness) of the DaC prompting.

Cons:
- As some reviewers point out, the study is relatively narrow in scope, and the proposed method only works (as the authors point out) on a very restricted set of tasks. How to reliably determine whether a task is applicable in practice also remains to be a question.